# Non-Convex Bilevel Games
# with Critical Point Selection Maps

**Michael Arbel and Julien Mairal**
Univ. Grenoble Alpes, Inria, CNRS, Grenoble INP, LJK, 38000 Grenoble, France
`firstname.lastname@inria.fr`

## Abstract

Bilevel optimization problems involve two nested objectives, where an upper-level objective depends on a solution to a lower-level problem. When the latter is non-convex, multiple critical points may be present, leading to an ambiguous definition of the problem. In this paper, we introduce a key ingredient for resolving this ambiguity through the concept of a *selection* map which allows one to choose a particular solution to the lower-level problem. Using such maps, we define a class of hierarchical games between two agents that resolve the ambiguity in bilevel problems. This new class of games requires introducing new analytical tools in Morse theory to extend *implicit differentiation*, a technique used in bilevel optimization resulting from the implicit function theorem. In particular, we establish the validity of such a method even when the latter theorem is inapplicable due to degenerate critical points. Finally, we show that algorithms for solving bilevel problems based on unrolled optimization solve these games up to approximation errors due to finite computational power. A simple correction to these algorithms is then proposed for removing these errors.

## 1 Introduction

Bilevel optimization has proven to be a major tool for solving machine learning problems that possess a nested structure such as hyper-parameter optimization [17], meta-learning [6], reinforcement learning [23, 33], or dictionary learning [38]. Introduced in the field of economic game theory in [49], a bilevel optimization problem can be understood as a game between a *leader* and a *follower* each of which optimizes their own objective function but where the leader can anticipate follower's actions. In the context of machine learning, the leader typically optimizes a hyper-parameter over a validation loss while the follower optimizes the model parameter on a training loss [37].

Bilevel optimization introduces many challenges. In particular, when multiple optimal solutions are available to the follower, the leader would need to optimize a different objective depending on the follower's strategy to select an optimal solution. As a result, the bilevel problem becomes ambiguously defined without knowing the follower's strategy [35]. A large body of work on bilevel programs for machine learning gets around these considerations by assuming the follower to have a unique optimal choice, a situation that typically occurs when the follower's objective is strongly convex, leading to efficient and scalable algorithms [1, 2, 7, 14, 20, 32, 33, 47]. However, in many machine learning applications, the strong convexity of the follower's objective is an unrealistic assumption. This is particularly the case in the context of deep learning, where the follower's objective, the training loss, can be highly non-convex in the parameters of the model and can have regions of flat optima due to symmetries and other degeneracies [15, 30].

In the literature on mathematical optimization, the ambiguity in bilevel problems is often resolved by making an additional assumption on the follower's strategy for choosing their optimal solution. In particular, two problems are often considered: *optimistic and pessimistic bilevel programs*, see [13].

36th Conference on Neural Information Processing Systems (NeurIPS 2022).

Both problems rely on two assumptions: (i) the follower is using a strategy for selecting a solution to their problem that is either improving or degrading the leader's objective and (ii) the leader knows exactly what strategy the follower is using. These assumptions are strong from a game-theoretical perspective and often unrealistic for machine learning problems such as hyper-parameter optimization. Still, optimistic/pessimistic bilevel games are well defined and early works have proposed several algorithms to solve them with strong convergence guarantees [55, 56, 57]. Yet, these algorithms are often ill-suited to large-scale and high-dimensional problems arising in machine learning applications as they rely on second-order optimization methods such as Newton's method [21]. For this reason, scalable first-order algorithms for such games have been proposed recently [34, 35].

However, many of the best-performing approaches for hyper-parameter optimization rely neither on an optimistic nor a pessimistic formulation of the bilevel problem [50]. Instead, they often rely on algorithms initially designed for bilevel problems with strongly convex lower objectives even though the convexity assumption does not hold [37]. Consequently, these algorithms are solving a seemingly ill-defined bilevel program due to the ambiguity in the way the follower selects their solution. However, their ability to provide models with good empirical performance raises the question of whether these algorithms are solving another class of well-defined hierarchical problems beyond optimistic and pessimistic bilevel programs that are still relevant for machine learning.

In this work, we answer the above question by introducing *Bilevel Games with Selection* (BGS), a class of games between two agents: a leader and a follower, where the leader uses a mechanism for anticipating the solution of the follower without knowing the exact follower's strategy. We define such a mechanism using the notion of a *selection*, which is simply a map for selecting a particular solution to the follower's objective given the current state of the game. In particular, BGS recovers a usual bilevel program when the follower's objective admits a unique solution. By playing a BGS, the agents seek an equilibrium point for which each of their objectives ceases to vary. The equilibria are completely determined by the *selection* thus resulting in a well-defined problem.

When the selection is differentiable, the equilibrium point can be characterized by a first-order optimality condition which enables gradient-based approximations. More precisely, we show that *implicit differentiation* [42], which, a priori, is only valid when the critical points of the follower's objective are non-degenerate, remains applicable for solving BGS even when these critical points are degenerate. To this end, we consider a general construction of the selection as the limit of a gradient flow of the follower's objective and prove the differentiability of such a selection near local minimizers, provided the follower's objective satisfies a generalization of the *Morse-Bott property* [4, 16]. We then characterize the differential of the selection as a solution to a linear system thus extending implicit differentiation to degenerate critical points. Finally, we leverage this characterization to show that popular algorithms based on iterative differentiation (ITD) [5] find fixed points approximating the BGS's equilibria up to approximation errors. We then introduce a simple corrective term to these algorithms based on implicit differentiation to remove these errors.

## 2 Related Work

**Iterative/Unrolled optimization (ITD)** is a class of methods approximating the lower-level solution map by a differentiable function obtained through successive gradient updates [5]. When the lower-level objective is strongly convex, these algorithms solve a well-defined bilevel problem up to an error that is controlled by increasing the computational budget for the approximate solution [25]. Our analysis suggests a simple algorithmic correction to these approaches which can result in solutions to a bilevel game with a constant budget for the approximate solution.

**Approximate Implicit Differentiation (AID)** is a class of methods approximating the variations of the lower-level solution map using the Implicit Function theorem [18, 19, 42, 43]. The non-degeneracy requirement under which the latter theorem holds restricts the applicability of AID to, essentially, strongly convex lower-level objectives. These algorithms admit fixed points that match the solutions to the bilevel problem [19, 23, 24, 25]. As such, they typically require a smaller computational budget than ITD [2, 25]. Recently, [8, 10, 9] extended AID to non-smooth objectives while still requiring non-degenerate critical points. The present work is complementary to these works as it extends AID to smooth objectives that have possibly degenerate critical points.

**Optimistic and pessimistic bilevel optimization.** When the lower-level objective is non-convex, the ambiguity of the problem arising from the multiplicity of the lower-level solutions can be re-

solved by optimizing the upper-level objective over all such possible solutions [53, 58]. The *optimistic* and *pessimistic* problems arise when either minimizing or maximizing the upper-level over all such lower-level solutions. Early works proposed to solve these problems using exact penalization [57], second-order optimization [55, 56] or smoothing method [54]. However, these approaches are hard to scale to the high dimensional problems arising in machine learning. More recently, [35, 34] considered first-order methods based on unrolled optimization or interior-point methods for solving optimistic bilevel problems and provided approximation guarantees. However, as shown in [50], most practical applications to bilevel optimization rely on a formulation that goes beyond optimistic or pessimistic formulations. The present work departs from these approaches and instead introduces a bilevel game that is more tractable to solve. We show that popular bilevel algorithms, such as unrolled optimization, yield approximations of these games.

## 3 Non-Convex Bilevel Optimization with Selection

**Notations.** Define $\mathcal{X} = \mathbb{R}^p$ and $\mathcal{Y} = \mathbb{R}^d$ for some positive integers $p$ and $d$. We consider two real valued functions $f$ and $g$ defined on $\mathcal{X} \times \mathcal{Y}$ and assume $g$ to be twice-continuously differentiable.

### 3.1 Background on Bilevel Optimization

A bilevel program is an optimization problem where an upper-level objective $f$ defined over a set $\mathcal{X} \times \mathcal{Y}$ of variables $(x, y)$ is optimized in the first variable $x$ under the constraint that the second variable $y$ is optimal for a lower-level objective $y \mapsto g(x, y)$ depending on the upper-variable $x$. When $g(x, .)$ admits a unique minimizer denoted by $y^\star(x)$, which is the case if $y \mapsto g(x, y)$ is strongly convex, the bilevel problem is well-defined and can be expressed as:

$$\min_{x \in \mathcal{X}} f(x, y^\star(x)), \qquad y^\star(x) := \arg\min_{y \in \mathcal{Y}} g(x, y). \tag{BP}$$

When $g$ is non-convex, the set of minimizers $T(x) := \arg\min_y g(x, y)$ may contain more than one element making (BP) ambiguous. A possible approach for resolving the ambiguity is to adopt a game-theoretical point of view, where a lower-level agent uses a particular strategy for selecting a solution in $T(x)$. For instance, in *pessimistic* bilevel games, the lower agent chooses a minimizer of $g(x, .)$ that maximizes $f(x, .)$ while the upper agent minimizes the resulting worst-case loss $F$ in $x$:

$$\textbf{(UL):} \quad \min_{x \in \mathcal{X}} F(x), \qquad \text{and} \quad \textbf{(LL):} \quad F(x) := \max_{y \in \mathcal{Y}} f(x, y) \quad \text{s.t.} \quad y \in T(x). \tag{pessimistic-BG}$$

Similarly, an *optimistic* bilevel game can be obtained by replacing maximization with minimization so that both agents cooperate. While these approaches are highly relevant from a game-theoretical point of view, many machine learning applications do not rely on a pessimistic/optimistic bilevel formulation. For instance, for hyper-parameter optimization, the lower agent may have access to training data, but it should not have access to the validation data processed (used in $f$) by the upper agent. Instead, a popular approach consists of applying algorithms designed for bilevel programs that admit unique solutions for the lower problems, even though this assumption may not hold in practice [37]. In the next section, we introduce a class of games that allow characterizing the equilibrium points obtained by these popular algorithms while resolving the ambiguity of non-convex bilevel problems and bypassing the limitations of pessimistic/optimistic bilevel formulations.

### 3.2 Bilevel Games with Selection (BGS)

We introduce a new class of nested games for bilevel optimization with two agents, a *leader* and a *follower*. The *follower* minimizes the lower-level objective $g$ w.r.t. a variable $y$ in $\mathcal{Y}$. Similarly, the *leader* minimizes the upper-level objective $f$ w.r.t. a variable $x \in \mathcal{X}$ while anticipating the *follower*'s solution. More precisely, the *leader* has access to a *selection map*: $\phi : \mathcal{X} \times \mathcal{Y} \to \mathcal{Y}$ to choose a unique critical point $\phi(x, y)$ of $y \mapsto g(x, y)$ given the current state of the game $(x, y) \in \mathcal{X} \times \mathcal{Y}$ thus allowing the leader to anticipate the follower's solution. Typically, the selection $\phi(x, y)$ represents the critical point that is *selected* by an optimization process of $g(x, .)$ starting from an initial condition $y$ (*e.g.*, the limit of a gradient flow for a gradient descent algorithm). The Bilevel Game with Selection (BGS) is therefore defined as the following interdependent optimization problems:

$$\textbf{(UL):} \quad \min_{x \in \mathcal{X}} \mathcal{L}_\phi(x, y) := f(x, \phi(x, y)), \qquad\qquad \textbf{(LL):} \quad \min_{y \in \mathcal{Y}} g(x, y). \tag{BGS}$$

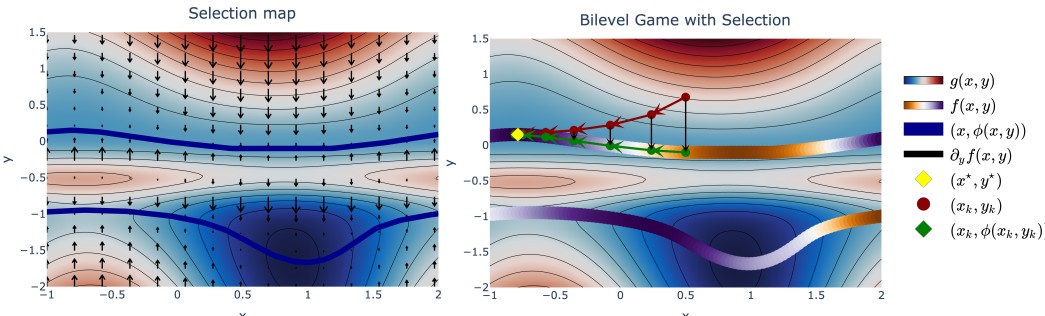

Figure 1: Left: Heatmap of the lower-level objective $g(x, y)$. The local minimizers of $y \mapsto g(x, y)$ are represented by the 'critical lines' in blue. The selection map $\phi(x, y)$ is defined by following the vector field $\partial_y g(x, y)$, in black. Right: Iterates $(x_k, y_k)$ (in red) obtained by playing a BGS. The follower finds the next update $y_k$ by optimizing $y \mapsto g(x_k, y)$ starting from previous iterate $y_{k-1}$. The leader finds the next update $x_k$ by optimizing the upper-level objective $f$ along the 'critical lines' (iterates in green).

Given a selection map $\phi$, the game (BGS) is well-defined and does not suffer from the ambiguity problem in (BP). The explicit dependence of $\phi(x, y)$ on the initialization $y$ might seem unnecessary at first, as one could simply fix $y$ to some value $y_0$ and consider only the dependence on the variable $x$. However, such a dependence on the variable $y$ allows performing *warm-start* [50], where the lower-level problem is optimized starting from a previous state of the game, thus resulting in computational savings Figure 1. We provide below a formal definition for the selection map.

**Definition 1** (**Selection map**). *Given a continuously differentiable function $g : \mathcal{X} \times \mathcal{Y} \to \mathbb{R}$, the map $\phi : \mathcal{X} \times \mathcal{Y} \to \mathcal{Y}$ is a selection if it satisfies the following properties for any pair $(x, y) \in \mathcal{X} \times \mathcal{Y}$:*

1. **Criticality:** *The element $y' = \phi(x, y)$ is a critical point of $g(x, .)$, i.e. $\partial_y g(x, y') = 0$.*

2. **Self-consistency:** *If $y$ is a critical point of $g(x, .)$ i.e. $\partial_y g(x, y) = 0$, then $\phi(x, y) = y$.*

*Criticality* ensures the leader possesses a hierarchical advantage in that they know what are the optimal choices accessible to the follower. *Self-consistency* implies that the leader makes a guess that is not contradicting the current choice $y$ of the follower. Both properties ensure the leader can rationally anticipate the follower's actions from the current state of the game $(x, y)$. We will see in Section 4, under mild assumptions on $g$, that it is always possible to define a selection $\phi$ as the limit of a continuous-time gradient flow of $y \mapsto g(x, y)$ initialized at $y$. Moreover, as we discuss later in Section 5, the selection does not need to be explicitly constructed for solving (BGS) in practice. It can be simply related to the implicit bias of the algorithm used for solving the follower's problem.

**Connection to (BP).** When the lower-level objective $y \mapsto g(x, y)$ admits a unique minimizer $y^\star(x)$, it is easy to check that there exists a unique selection map $\phi$ satisfies $\phi(x, y) = y^\star(x)$. Hence, (BGS) recovers the bilevel problem in (BP) as a particular case.

**Connection to (pessimistic-BG) or the optimistic variant.** Key differences between (BGS) and pessimistic or optimistic games is that (i) the follower has never access to the upper function $f$ with (BGS), which matches practical hyper-parameter optimization applications where $f$ relies on a validation dataset, whereas $g$ relies on a distinct training set; (ii) the leader in (pessimistic-BG) does not take into account the strategy used by the follower, whereas the leader in (BGS) makes more rational choices by guessing the strategy of the follower through the selection map $\phi$.

**First-order equilibrium conditions.** The agents can play the game (BGS) by successively taking actions $(x_k, y_k)$ to improve their own objectives $x \mapsto \mathcal{L}_\phi(x, y_{k-1})$ and $y \mapsto g(x_k, y)$, by hoping the strategy will reach an equilibrium pair $(x^*, y^*)$ Figure 1(Right). In the case where $f$, $g$ and $\phi$ are differentiable at $(x^*, y^*)$, the equilibrium pair is characterized by a first-order stationary condition:

$$\partial_x \mathcal{L}_\phi(x^\star, y^\star) = \partial_x f(x^\star, y^\star) + \partial_x \phi(x^\star, y^\star) \partial_y f(x^\star, y^\star) = 0, \qquad \partial_y g(x^\star, y^\star) = 0. \qquad \text{(SC)}$$

When $g$ is smooth and strongly convex in $y$, the implicit function theorem [28, Theorem 5.9] ensures that $\phi$ is differentiable and provides an expression of $\partial_x \phi(x^\star, y^\star)$ as a solution to a linear system which key for implicit differentiation. This allows to devise efficient algorithms using estimates of the gradient $\partial_x \mathcal{L}_\phi$, see, *e.g.*, [2]. However, extensions of the implicit function theorem, such as the

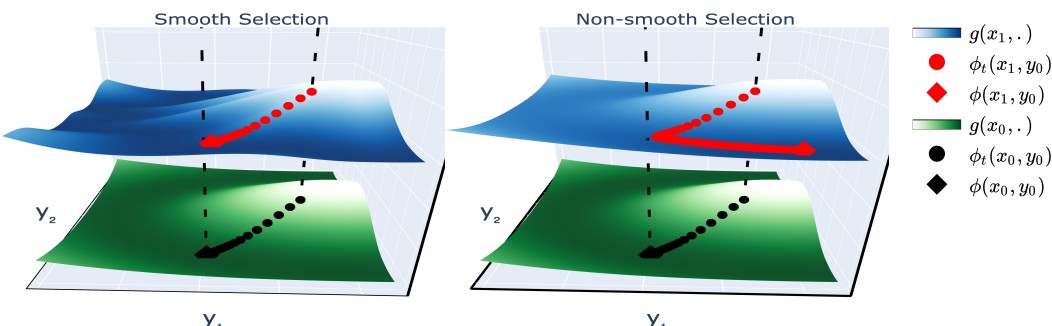

Figure 2: Two examples of functions $g$ with different behaviors of the gradient flow under perturbations of $x$. In both figures, the green surface represents a function $y \mapsto g(x_0, y)$ with $y \in \mathbb{R}^2$ resembling a *Mexican hat* which has a manifold of (degenerate) local minimizers (in dark green). The blue surfaces represent *deformed* versions of the Mexican hat function when the parameter $x$ is slightly perturbed $x_1 \approx x_0$. Depending on the deformation, the resulting function $y \mapsto g(x_1, y)$ can either preserve the same type of critical points as the unperturbed function, i.e. local minimizers remain local minimizers (Left), or change their type, i.e.: local minimizers can become saddle-points (Right). Left: the selection behaves smoothly as a function of the deformation. Right: the selection is discontinuous since the gradient flow is pushed away from $\phi(x_0, y_0)$ which is deformed into a saddle point.

*constant rank theorem* [29, Theorem 4.12], for cases where $g$ has possibly degenerate critical points require strong assumptions on $g$ which are unrealistic in machine learning. In the next section, we provide new analytical tools for extending *implicit differentiation* by studying the differentiability of a family of selection maps corresponding to a large class of functions $g$. The resulting expression will be key for devising first-order methods to solve (BGS), as discussed in Section 5.

## 4    Selection Based on Gradient Flows for Parameteric Morse-Bott Functions

In this section, we extend implicit differentiation to a class of functions with possibly degenerate critical points. To this end, we consider a particular selection $\phi(x, y)$ obtained as the limit of a gradient flow $(\phi_t(x, y))_{t \geq 0}$ of $g(x, .)$ initialized at $y$. We then study the *differentiability* w.r.t. $x$ of the selection by analyzing the dynamics of such a gradient flow. For general non-convex functions, the selection might be non-differentiable since a small perturbation to the parameter $x$ can change the geometry of the critical points of $g$, causing the perturbed flow to move away from the non-perturbed one (see Figure 2). We are therefore interested in functions $g$ preserving the local geometry near critical points as $x$ varies. In Section 4.1, we introduce such a class of functions called parametric Morse-Bott functions, which covers many practical machine learning models. We then show, in Section 4.2, that the selection resulting from such a function is differentiable near local minima.

### 4.1    Parameteric Morse-Bott Functions

We introduce parametric Morse-Bott functions, a class of parametric functions $g : \mathcal{X} \times \mathcal{Y} \to \mathbb{R}$ with parameter $x$ in $\mathcal{X}$ extending the more familiar notion of Morse-Bott functions (Appendix A.1, [16]) to account for the effect of the parameter $x$ on the geometry of critical points.

**Definition 2 (Parametric Morse-Bott function.).** *Let $g : \mathcal{X} \times \mathcal{Y}$ be a real-valued twice continuously differentiable function and define the set of* augmented critical points $\mathcal{M}$ *as follows:*

$$\mathcal{M} := \{(x, y) \in \mathcal{X} \times \mathcal{Y} \mid \partial_y g(x, y) = 0\} \tag{1}$$

*Let $(x_0, y_0) \in \mathcal{M}$. We say that $g$ is Morse-Bott at $y_0$ w.r.t. $x_0$, if there exists an open neighborhood $\mathcal{V}$ of $(x_0, y_0)$ s.t. the intersection $\mathcal{M} \cap \mathcal{V}$ is a $C^2$-connected sub-manifold of $\mathcal{X} \times \mathcal{Y}$ of dimension:*

$$\dim(\mathcal{M} \cap \mathcal{V}) = \dim(\mathcal{X}) + \dim(Ker(\partial^2_{yy} g(x_0, y_0))).$$

*$g$ is a parametric Morse-Bott function if for any $(x_0, y_0) \in \mathcal{M}$, $g$ is Morse-Bott at $y_0$ w.r.t. $x_0$.*

The functions in Definition 2 satisfy a condition that is stronger than simply satisfying the Morse-Bott property at any parameter value $x$ (Definition 3 of Appendix A.1). Indeed, we show in Proposition 7 of Appendix A.2 that, for any $x_0 \in \mathcal{X}$, the function $y \mapsto g(x_0, y)$ is a Morse-Bott function,

meaning that the critical set $C(x_0)$ of $y \mapsto g(x_0, .)$ near a critical point $y_0$ is locally a $C^2$ connected sub-manifold of $\mathcal{Y}$ of dimension equal to the dimension of the null-space of the Hessian $\partial_{yy}^2 g(x_0, y_0)$. For conciseness, we introduce the following assumption which ensures $g$ satisfies the condition of Definition 2 as well as possesses continuous third-order derivatives.

**Assumption 1** (**Parameteric Morse-Bott property**). *The function $g$ is at least three-times continuously differentiable and is a parameteric Morse-Bott function as defined in Definition 2.*

**Examples of parametric Morse-Bott function.** A notable class of parametric Morse-Bott functions is the one containing all twice-continuously differentiable functions that are strongly convex or, more generally, possess only non-degenerate critical points in the second variable as shown in Proposition 8 of Appendix A.2. Note that parametric Morse-Bott functions need not be convex and can have multiple (possibly degenerate) local minima, saddle-points, and local maxima.

Another class of functions, this time with possibly degenerate critical points, are those that can be expressed as a composition of some Morse-Bott function $h$ and a family $(\tau_x)_{x \in \mathcal{X}}$ of diffeomorphisms on $\mathcal{Y}$ parameterized by $x$, i.e. $g(x, y) = h(\tau_x(y))$. This particular form is relevant in generative modeling where the diffeomorphisms are defined using normalizing flows of parameter $x$ [44].

The condition in Definition 2 ensures that the degree of freedom of the augmented critical set $\mathcal{M}$ is exactly determined by the degree of freedom of the parameter $x$ and the degree of degeneracy of the Hessian at a critical point $y$. This condition is precisely what guarantees the stability of the local shape of critical points when the parameter $x$ varies as we formalize through the next theorem.

**Theorem 1** (**Morse-Bott lemma with parameters**). *Let $g$ be a function satisfying Assumption 1. Let $(x_0, y_0)$ in $\mathcal{M}$ be an augmented critical point of $g$. Denote by $\mathcal{K}$ the null space of the Hessian $A_0 := \partial_{yy}^2 g(x_0, y_0)$ and by $\mathcal{K}^\perp$ its orthogonal complement in $\mathcal{Y}$. Let $J_0$ be a diagonal matrix with diagonal element given by the sign of the non-zero eigenvalues of $A_0$. Then, there exists open neighborhoods $\mathcal{U}$ and $\mathcal{V}$ of $(x_0, 0_\mathcal{K}, 0_{\mathcal{K}^\perp})$ and $(x_0, y_0)$ in $\mathcal{X} \times \mathcal{K} \times \mathcal{K}^\perp$ and $\mathcal{X} \times \mathcal{Y}$, and a diffeomorphism $\psi : \mathcal{U} \to \mathcal{V}$ preserving the first variable, i.e. $\psi(x, r, w) = (x, y)$ for any $(x, r, w) \in \mathcal{U}$, with $\psi(x_0, 0_\mathcal{K}, 0_{\mathcal{K}^\perp}) = (x_0, y_0)$ such that $g$ admits the representation:*

$$g(\psi(x, r, w)) = g(\psi(x, 0_\mathcal{K}, 0_{\mathcal{K}^\perp})) + \frac{1}{2} w^\top J_0 w, \qquad \forall (x, r, w) \in \mathcal{U}.$$

Theorem 1, which is proven in Appendix A.3, shows that, near an augmented critical point $(x_0, y_0)$, $g$ looks like a quadratic function up to an additive term that depends only on the parameter $x$. Moreover, slightly varying the parameter $x$ does not change the quadratic function and thus preserves the local shape near critical points. Theorem 1 is an extension of the *Morse-Bott lemma* [16, Theorem 2.10] to the case when there is a dependence on a parameter $x$. It can also be seen as an extension of the *Morse lemma with parameters* [16, Theorem 4] which allows dependence to a parameter $x$ but requires the critical points to be non-degenerate (invertible matrix $A_0$). To our knowledge, Theorem 1 is the first result in the literature providing a decomposition of parametric functions with degenerate critical points into the sum of a quadratic non-degenerate term and a singular term depending only on the parameter $x$. We present now a corollary of Theorem 1 which is a strengthened version of the standard Łojasiewicz inequality [36] that will be essential for our subsequent analysis.

**Proposition 1** (**Locally Uniform Łojasiewicz gradient inequality**). *Let $g$ be a function satisfying Assumption 1 and let $(x_0, y_0)$ be in $\mathcal{M}$ the augmented critical set defined in Definition 2. Then, there exists an open neighborhood $\mathcal{U}$ of $(x_0, y_0)$ and a positive number $\mu > 0$ such that $y \mapsto g(x, y)$ is constant on the set $\mathcal{M} \cap \mathcal{U}$ with some common value $G(x) := g(x, y)$ and the following holds:*

$$\mu |g(x, y) - G(x)| \leq \frac{1}{2} \|\partial_y g(x, y)\|^2, \qquad \forall (x, y) \in \mathcal{U}.$$

Proposition 1, which is proven in Appendix A.3, ensures that the Łojasiewicz gradient inequality holds uniformly on $(x, y)$ near any augmented critical point $(x_0, y_0)$. This result will be essential in Section 4.2 for defining a selection $\phi$ obtained as limits of gradient flows and to obtain a locally uniform control of these flows in the parameter $x$. This in turn will allow us to obtain the differentiability of the selection in the parameter $x$ whenever $\phi(x, y)$ is a local minimum.

## 4.2 Smoothness of Selections Based on Gradient Flows of a Parametric Morse-Bott Function

We consider a construction for the selection $\phi$ in Definition 1 as a limit of a continuous-time gradient flow of $g$. More precisely, we define a continuous-time trajectory $(\phi_t(x, y))_{t \geq 0}$ in $\mathcal{Y}$ initialized at

$\phi_0(x, y) = y$ and driven by the differential equation:

$$\frac{d\phi_t(x, y)}{dt} = -\partial_y g(x, \phi_t(x, y)). \tag{GF}$$

Provided $\phi_t(x, y)$ converges towards some element $\phi(x, y)$ as $t \to +\infty$, we can expect such a limit to satisfy both conditions of Definition 1, therefore constituting a valid selection. However, for general non-convex functions, $\phi_t(x, y)$ might not always converge [36]. To guarantee the existence and convergence of the flow, we make the following assumptions on the function $g$.

**Assumption 2 (Smoothness).** *There exists $L > 0$ such that $y \mapsto \partial_y g(x, y)$ is L-Lipschitz for any $x \in \mathcal{X}$.*

**Assumption 3 (Coercivity).** *For any $x \in \mathcal{X}$, it holds that $g(x, y) \to +\infty$ as $\|y\| \to +\infty$.*

The smoothness assumption in Assumption 2 is standard and guarantees the existence of the flow by the Cauchy-Lipschitz theorem. The coercivity condition in Assumption 3 guarantees that $\phi_t(x, y)$ cannot escape to infinity. It can be easily enforced by adding a small $\ell_2$-penalty to a non-negative loss (such as cross-entropy or mean-squared loss) which is already a common practice in machine learning. These assumptions, along with Assumption 1 ensure that the limit $\phi(x, y)$ always exists as we summarize in the following proposition, which is proven in Appendix B.

**Proposition 2.** *Under Assumptions 1 to 3, and for any $(x, y) \in \mathcal{X} \times \mathcal{Y}$, the gradient flow (GF) always converges towards a critical point $\phi(x, y)$ of $y \mapsto g(x, y)$ and the map $(x, y) \mapsto \phi(x, y)$ is a selection map as defined in Definition 1. We call $\phi$ the* flow selection *relatively to $g$.*

Proposition 2 is a consequence of a general result that holds for functions satisfying a Łojasiewicz gradient inequality [3, 40] which is the case here by Proposition 1. From now on, we restrict our attention to the selection $\phi$ defined in Proposition 2. Even though $\phi$ satisfies the implicit equation $\partial_y g(x, \phi(x, y)) = 0$, we cannot rely anymore on the implicit function theorem for studying the differentiability of $\phi(x, y)$ in $x$ since $g$ can have degenerate critical points. Instead, we propose to characterize the differentiability of $\phi$ by studying the limit of $U_t(x, y) := \partial_x \phi_t(x, y)$ which is formally driven by a linear differential equation of the form:

$$-\frac{dU_t(x, y)}{dt} = \partial_{xy}^2 g(x, \phi_t(x, y)) + U_t(x, y) \partial_{yy}^2 g(x, \phi_t(x, y)). \tag{2}$$

Had we known in advance that $\phi(x, y)$ is differentiable in $x$, the limit $U_\infty(x, y)$ of $U_t(x, y)$ as $t \to +\infty$, whenever defined, would be a promising candidate for the differential of $\phi(x, y)$ in $x$. Such a limit is indeed expected to satisfy the following linear equation:

$$0 = \partial_{xy}^2 g(x, \phi(x, y)) + U_\infty(x, y) \partial_{yy}^2 g(x, \phi(x, y)). \tag{3}$$

A first challenge is to ensure that $U_t$ does not diverge. For critical points $\phi(x, y)$ that are not local minima, it is easy to see that the Hessian $\partial_{yy}^2 g(x, \phi_t(x, y))$ must have a negative eigenvalue for $t$ large enough, therefore causing the system (2) to diverge. Intuitively, unless $\phi(x, y)$ is a local minimum, there is no reason to expect $\phi(x, y)$ to be differentiable or even continuous in $x$, simply because $\phi(x, y)$ would be an unstable fixed-point of the flow $\phi_t(x, y)$, so that any change in $x$ might cause a large variation in $\phi(x, y)$. The possible non-differentiability of $\phi(x, y)$ for critical points that are not local minima is not problematic in practice, since for almost all initial conditions $y$ of the flow $\phi_t(x, y)$, the limit $\phi(x, y)$ is guaranteed to be a local minimizer [41]. In addition, we show in Proposition 13 of Appendix B.3 that if $\phi(x_0, y)$ is a local minimum, then $\phi(x, y)$ must also be a local minimum in a neighborhood of $x_0$.

Nevertheless, even for local minima, if the Hessian $\partial_{yy}^2 g(x, \phi(x, y))$ is non-invertible, (3) might never hold if $\partial_{xy}^2 g(x, \phi(x, y))$ does not belong to the image of the Hessian. However, we show in Proposition 6 of Appendix A.2 that, for any pair $(x, y)$ of critical points, $\partial_{xy}^2 g(x, y)$ must always belong to the span of the Hessian $\partial_{yy}^2 g(x, y)$ as soon as $g$ satisfies Assumption 1, therefore ensuring that (3) admits a solution. The following theorem, which is proven in Appendix C, establishes the differentiability of $\phi$ at local minima and shows that $\partial_x \phi$ is exactly given by the limit $U_\infty$.

**Theorem 2 (Degenerate implicit differentiation.).** *Let $g$ be a function satisfying Assumptions 1 to 3 so that the flow selection $\phi$ is well-defined. Let $(x_0, y_0)$ be in $\mathcal{X} \times \mathcal{Y}$. If $\phi(x_0, y_0)$ is a local minimizer of $y \mapsto g(x_0, y)$, then there exists a neighborhood $\mathcal{U}$ of $x_0$ on which $x \mapsto \phi(x, y_0)$ is differentiable with differential $\partial_x \phi(x, y_0) = U_\infty(x, y_0)$. Moreover, if $y_0$ is a local minimizer of $y \mapsto g(x_0, y)$, then, denoting by $\dagger$ the pseudo inverse operator, $\partial_x \phi(x_0, y_0)$ is exactly given by:*

$$\partial_x \phi(x_0, y_0) = -\partial_{xy} g(x_0, y_0)(\partial_{yy} g(x_0, y_0))^\dagger. \tag{4}$$

The expression in (4) is very similar to the one that would arise by application of the implicit function theorem to a strongly convex function $g$. However, the proof technique does not rely on such a theorem which would not be applicable here. The key technical challenges in proving the above result are: (i) showing that $\phi(x,y)$ must be continuous at $x_0$ and (ii) controlling the error $\|U_t(x,y) - U_\infty(x,y)\|$ locally uniformly in $x$. The result follows by the application of classical uniform convergence results [46, Theorem 7.17]. The continuity of $\phi$ is established in Proposition 12 of Appendix B.3 and relies on a stability analysis of the flow $\phi_t$ performed in Appendix B.2. The uniform convergence of $U_t$ towards $U_\infty$ is shown in Proposition 17 of Appendix C and relies on a local uniform convergence of the flow $\phi_t$ towards $\phi$ which is proven in Proposition 14 of Appendix B.4. It is worth noting that, even though we identified $\partial_x \phi$ to be $U_\infty$, the latter is not fully characterized by (3) as it might contain a non-zero component in the null-space of the Hessian. However, when $(x_0, y_0)$ is an augmented critical pair of $g$, such a component vanishes, and $\partial_x \phi(x_0, y_0)$ is exactly determined by the minimal norm solution in (4). The latter fact has practical implications when designing algorithms for solving (BGS) as we discuss next.

## 5 Algorithms

### 5.1 Unrolled Optimization for BGS

Unrolled optimization constructs a map $\varphi_T(x,y)$ approximating a critical point of the function $y \mapsto g(x,y)$ for any fixed $x$ by applying a finite number $T > 0$ of gradient updates starting from some initial condition $y$. By convention, we set $\varphi_0(x,y)=y$. Hence, $\varphi_T$ can be understood as an approximation to the selection map defined in Section 4.2. We emphasize that $\varphi_T$ is not a selection (Definition 1) since $\varphi_T(x,y)$ is not a critical point of $g$ in general. Nevertheless, it provides a tractable approximation to critical points which is key for constructing practical algorithms for bilevel optimization. The gradient of $\varphi_T(x,y)$ w.r.t. $x$ is then obtained by differentiating through the optimization steps and used to optimize the approximate upper-level objective:

$$\mathcal{L}_T(x,y) := f(x, \varphi_T(x,y)).$$

Given the $k$-th upper-level iterate $x_k$ and an initial condition $\tilde{y}_k$ for the unrolled optimization, these approaches compute an approximation $y_k=\varphi_T(x_{k-1}, \tilde{y}_k)$ and find an update direction $d_k$ for the upper-level variable $x$ by differentiating $\mathcal{L}_T(x, \tilde{y}_k)$ in $x$ at the current iterate $x_{k-1}$. The following iterate $x_k$ is obtained by applying an update procedure, such as $x_k=x_{k-1}-\gamma d_k$ for positive small enough step-size $\gamma$. In Algorithm 1, we present several variants of these schemes, including a simple correction allowing them to solve (BGS) instead of an approximation.

The initial condition $\tilde{y}_k$ is often computed using a warm-start procedure $\tilde{y}_k=\mathcal{I}_M(x_{k-1}, y_{k-1})$. The simplest procedure is to set $\tilde{y}_k=y_{k-1}$ in which case $\mathcal{I}_0(x,y)=y$. However, it is not uncommon to perform $M>0$ optimization steps to minimize the objective $y \mapsto g(x_{k-1}, y)$ starting from $y_{k-1}$. By doing so, gradient unrolling stops at $\tilde{y}_k$ and ignores the dependence of $\tilde{y}_k$ on $y_{k-1}$, resulting in Truncated unrolled optimization [47]. Algorithm 1 summarizes these approaches when the binary variable **AddCorrection** is set to **False**. To characterize the limit points of Algorithm 1, we make the following assumptions on $\mathcal{I}_M, \varphi_T$.

---

**Algorithm 1** BGS-Opt$(x_0, y_0)$

1: Inputs: $x_0, y_0$,
2: Parameters: $K, T, M, \gamma$ **AddCorrection**
3: **for** $k \in \{1, ..., K+1\}$ **do**
4:     $\tilde{y}_k \leftarrow \mathcal{I}_M(x_{k-1}, y_{k-1})$. # Warm-start.
5:     $y_k \leftarrow \varphi_T(x_{k-1}, \tilde{y}_k)$ # Unrolled optimization.
6:     $d_k \leftarrow \partial_x \mathcal{L}_T(x_{k-1}, \tilde{y}_k)$
7:     **if AddCorrection= True then**
8:         $v_k \leftarrow \partial_y \mathcal{L}_T(x_{k-1}, \tilde{y}_k)$
9:         $\xi_k \approx -(\partial_{yy} g(x_{k-1}, y_k))^\dagger v_k$ # Approx. solver
10:         $d_k \leftarrow d_k + \partial_{xy} g(x_{k-1}, y_k) \xi_k$ # Grad. correction
11:     **end if**
12:     $x_k \leftarrow x_{k-1} - \gamma d_k$ # Updating $x$
13: **end for**
14: Return $(x_K, y_K)$.

---

**Assumption 4.** *For any non-negative integers $M, T \geq 0$, the maps $\mathcal{I}_M$ and $\varphi_T$ are continuous on $\mathcal{X} \times \mathcal{Y}$ and take values in $\mathcal{Y}$, with $\varphi_T$ being continuously differentiable. Moreover, for any $(x,y) \in \mathcal{X} \times \mathcal{Y}$ s.t. $\partial_y g(x,y)=0$ and $M, T \geq 0$, there exists a matrix $D$ such that:*

$$\mathcal{I}_M(x,y) = \varphi_T(x,y) = y, \quad \partial_x \varphi_T(x,y) = \partial^2_{xy} g(x,y) D, \quad \partial_y \varphi_T(x,y) = I + \partial^2_{yy} g(x,y) D.$$

*Finally, for any $(x, y) \in \mathcal{X} \times \mathcal{Y}$, and $M, T \geq 0$ s.t. $T + M > 0$, the equality $y = \varphi_T(x, \mathcal{I}_M(x, y))$ implies that $y$ is a critical point of $g$, i.e. $\partial_y g(x, y) = 0$.*

**Assumption 5.** *$\varphi_T$ converges to a selection $\phi$ and $\partial_x \varphi_T$ converges uniformly near local minima.*

Assumption 4 is satisfied by many mappings used in practice such as $T$-steps of the gradient descent or proximal point algorithms, whenever $g$ is twice-continuously differentiable and $L$-smooth as shown in Proposition 19 of Appendix D. Assumption 5 is a discrete-time version of the uniform convergence result in Proposition 17 of Appendix C but that we directly assume here for simplicity. Under these assumptions we show that Algorithm 1 can find equilibria of (BGS) up to an approximation error resulting from the fact that $\varphi_T$ is not an exact selection.

**Proposition 3.** *Let $M, T$ be non-negative numbers s.t. $M + T > 0$ and let $(x_k, y_k)$ be the iterates of Algorithm 1 using the maps $\mathcal{I}_M$ and $\varphi_T$ and without any correction, i.e. **AddCorrection**=**False**. If $(x_k, y_k)$ converges to a limit point $(x_T^\star, y_T^\star)$ then, under Assumption 4:*

$$\partial_x \mathcal{L}_T(x_T^\star, y_T^\star) = 0, \qquad \partial_y g(x_T^\star, y_T^\star) = 0.$$

*Let $E$ be the set of limit points $(x_T^\star, y_T^\star)$ for $T \geq 0$. If $E$ is bounded and $y_T^\star$ is a local minimum of $g(x_T^\star, .)$ for any $T \geq 0$, then, under Assumptions 4 and 5, the elements of $E$ are approximate equilibria for (BGS):*

$$\limsup_T \|\partial_x \mathcal{L}_\phi(x_T^\star, y_T^\star)\| = 0, \qquad \partial_y g(x_T^\star, y_T^\star) = 0, \quad (\forall T > 0).$$

Proposition 3 shows that unrolled optimization algorithms approximately solve (BGS) in the limit where the number of unrolling steps $T$ of the $\varphi_T$ goes to infinity. This result is consistent with the ones obtained in [25] for the case where $g$ is strongly convex and illustrates the high computational cost for solving (BGS) without correcting for the bias introduced by unrolling. Next, we show how to get rid of such a bias in light of Theorem 2.

## 5.2   Implicit Gradient Correction

We propose to correct the bias of unrolling by exploiting the expression of the gradient $\partial_x \phi$ provided in Theorem 2. The key idea is to obtain an expression for $\partial_x \mathcal{L}_\phi(x, y)$ in terms of $\mathcal{L}_T$ and the second-order derivatives of $g$ which holds for any local minimizer $y$ of $y \mapsto g(x, y)$ as shown by the proposition below.

**Proposition 4.** *Let $\phi$ be the selection defined in Section 4.2 and $(x, y) \in \mathcal{X} \times \mathcal{Y}$ be s.t. $y$ is a local minimum of $y \mapsto g(x, y)$. Then, under Assumptions 1 to 4, $\partial_x \mathcal{L}_\phi(x, y)$ is given by the equation:*

$$\partial_x \mathcal{L}_\phi(x, y) := \partial_x \mathcal{L}_T(x, y) - \partial_{xy}^2 g(x, y) \big(\partial_{yy}^2 g(x, y)\big)^\dagger \partial_y \mathcal{L}_T(x, y).$$

Proposition 4, which is proven in Appendix D, suggests a simple correction for the gradient estimate $d_k$ in Algorithm 1. By doing so, the corrected algorithm would be performing an approximate gradient descent on each of the upper-level and lower-level objectives, suggesting that the algorithm may recover equilibrium points of (BGS) without having to increase the computation budget for the unrolling as we show later in Proposition 5. A simple way to proceed would to compute $c_k$ satisfying the approximate equation $c_k \approx - B_k(A_k)^\dagger v_k$, where $A_k = \partial_{yy}^2 g(x_{k-1}, y_k)$, $B_k := \partial_{xy}^2 g(x_{k-1}, y_k)$ and $v_k = \partial_y \mathcal{L}_T(x_{k-1}, \tilde{y}_k)$. More concretely, $c_k$ can be computed by setting $c_k = B_k \xi_k$ where $\xi_k$ approximates the minimum norm solution to the least squares problem:

$$\xi_k \approx \arg\min_\xi \|\xi\|^2, \qquad s.t. \quad \xi \in \arg\min_\xi \|A_k \xi + v_k\|^2, \tag{5}$$

**Approximate solution to (5).** It is possible to solve (5) approximately using an iterative procedure by constructing $N$ iterates $\xi^t$ starting from $\xi^0 = 0$ and performing (conjugate) gradient descent on the quadratic objective. This can be implemented efficiently using only Hessian vector products with the Hessian $A_k$ [37]. The constrained problem (5) can also be expressed as an unconstrained one by re-parametrizing $\xi = A_k z$:

$$\xi_k \approx A_k z_k^\star, \qquad s.t. \quad z_k^\star \in \arg\min_z \|A_k^2 z + v_k\|^2. \tag{6}$$

Eq. (6) has the advantage that $z_k^\star$ solves an unconstrained problem. As such, it is more amenable to applying a warm-start strategy, which can yield efficient approximation $z_k$ to $z_k^\star$ by exploiting previously computed approximation $z_{k-1}$ to $z_{k-1}^\star$ [2]. This strategy can be achieved using a standard

iterative algorithm $\mathcal{P}$ for approximately solving the least-squares problems, such as a fixed number of conjugate gradient iterations, that takes as input the matrix $A_k$, vector $v_k$ and initialization $z_{k-1} \approx z_{k-1}^\star$ and returns the next iterate $z_k \approx z_k^\star$. More formally we view $\mathcal{P}$ as a continuous map of $(A, v, z) \mapsto \mathcal{P}(A, v, z)$ returning a vector $z'$ and such that the only fixed points are exact solutions to the least square problem $\min_z \left\| A^2 z + v \right\|^2$. We refer to Appendix D.1 for examples of such maps. We can then define the iterates $z_k$ and $\xi_k$ as follows:

$$\xi_k = A_k z_k, \qquad z_k = \mathcal{P}(A_k, v_k, z_{k-1}). \tag{7}$$

The corrected algorithm is obtained by setting the variable **AddCorrection**=**True** in Algorithm 1 and computing the $\xi_k$ using any approximate solver including, in particular, the ones based on a warm-start strategy as in (7). The following proposition, with proof in Appendix D, shows that the proposed correction indeed yields equilibrium points of (BGS).

**Proposition 5.** *Let $(x_k, y_k)$ be the iterates obtained using Algorithm 1 with **AddCorrection**=**True** and $T + M > 0$ and assume that $\xi_k$ are computed using (7). If $(x_k, y_k, z_k)_{k \geq 0}$ converges to a limit point $(x^\star, y^\star, z^\star)$, then $y^\star$ is a critical point of $y \mapsto g(x^\star, y)$ and if, in addition, $y^\star$ is a local minimizer, then $(x^*, y^*)$ must be an equilibrium of (BGS) satisfying (SC):*

$$\partial_x \mathcal{L}_\phi(x^\star, y^\star) = 0 \quad \text{and} \quad \partial_y g(x^\star, y^\star) = 0$$

Proposition 5 shows that the proposed correction allows to recover equilibria of (BGS) without having to increase the number of iterations $T$ of the unrolled algorithm. This is by contrast with Proposition 3 where $T$ must increase to infinity, which would be impractical. We discuss in Appendix D.2 how different choices for the parameters $T$ and $M$ recover known algorithms. In particular, that Algorithm 1 with correction allows interpolating between two families of algorithms: (ITD) and (AID) while still recovering the correct equilibria. Numerical results illustrating the benefits of the correction are presented in Appendix E.

## 6 Discussion

We have introduced a bilevel game that resolves the ambiguity in bilevel optimization with non-convex objectives using the notion of selection maps. We have shown that many algorithms for bilevel optimization approximately solve these games up to a bias due to finite computational power. Our study of the differentiability properties of the selection maps has resulted in practical procedures for correcting such a bias and required the development of new analytical tools. This study opens the way for several avenues of research to understand the tradeoff between unrolling and implicit gradient correction for designing efficient algorithms. In future work, studying these algorithms in a non-smooth and stochastic setting would also be of great theoretical and practical interest.

**Funding** This project was supported by ANR 3IA MIAI@Grenoble Alpes (ANR-19-P3IA-0003).

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
