# A Morse-Bott Lemma with Parameters

## A.1 Background on Morse-Bott Functions

We recall the definition of classical Morse-Bott functions [4, 16], which we extend in Section 4.1 to the case where there is a dependence on some additional parameter $x$ in $\mathcal{X}$.

**Definition 3** (**Morse-Bott function**). *Let $h : \mathcal{Y} \to \mathbb{R}$ be a real-valued twice continuousely differentiable function. Define $\mathcal{C}_h$ to be the set of critical points of $h$ and consider $y_0 \in C_h$. We say that $h$ is Morse-Bott at $y_0$, if there exists a open neighborhood $\mathcal{V}$ of $y_0$ such that $C_h \cap \mathcal{V}$ is a connected sub-manifold of $\mathcal{Y}$ of dimension $\dim\left(Ker(\partial_{yy}^2 h(y_0))\right)$. We say that $h$ is a Morse-Bott function if for any $y_0 \in \mathcal{C}_h$, $h$ is Morse-Bott at $y_0$.*

Morse-Bott functions were introduced in the context of differential topology to analyze the geometry of a manifold by studying the properties of differentiable functions defined on that manifold [4]. Their main property is that all their critical points that are connected have the same type (same number of positive and negative eigenvalues for the Hessian), a fact expressed by the Morse-Bott lemma [16, Theorem 2.10] that we generalize to the parametric setting in Theorem 1. Morse-Bott functions form a *generic* class of functions [39], meaning that any smooth function can always be slightly perturbed to become a smooth Morse-Bott function. Hence, in principle, requiring that $y \mapsto g(x, y)$ is a Morse-Bott function for any parameter $x \in \mathcal{X}$ is essentially a mild assumption. The Morse-Bott property allows characterizing the geometry of critical points of $g(x, .)$ for any $x$ and ensures that the selection map $\phi$ is well-defined [11, Chapter 15]. However, this condition does not provide any information about how the set of critical points evolves as the parameter $x$ varies, which is crucial for the study of smoothness of the selection $\phi$. This is precisely why we introduced *parametric Morse-Bott functions* in Section 4.1.

## A.2 Properties of Parameteric Morse-Bott Functions.

In this section, we describe some elementary properties of parametric Morse-Bott functions. In particular, Proposition 6 shows that $\partial_{xy}^2 g(x, y)$ belongs to the range of $\partial_{yy}^2 g(x, y)$ whenever $(x, y)$ is an augmented critical point of $g$, i.e. $\partial_y g(x, y) = 0$. Proposition 7 shows that any parametric Morse-Bott function $g$ satisfies a pointwise Morse-Bott property in the sense of Definition 3. Finally, Proposition 8 and Lemma 1 provide examples of functions that satisfy the parametric Morse-Bott property. Recall $\mathcal{M}$ the set of augmented critical points of $g$:

$$\mathcal{M} = \{(x, y) \in \mathcal{X} \times \mathcal{Y} | \partial_y g(x, y) = 0\}. \tag{8}$$

**Proposition 6** (**Exact least square solution**). *Let $g$ be a parametric Morse-Bott function. Let $(x_0, y_0)$ be an element in $\mathcal{M}$ defined in (8) and define the matrices $A := \partial_{yy}^2 g(x_0, y_0)$ and $B := \partial_{xy}^2 g(x_0, y_0)$. Then, $B$ is in the range of $A$, i.e. there exists a matrix $U$ such that $B=UA$.*

*Proof.* Recall that $\mathcal{M}$ is the set of augmented critical points of $g$. Since $g$ is a parametric Morse-Bott function, there exists a neighborhood $\mathcal{U}$ of $(x_0, y_0)$ such that the augmented critical set $\mathcal{M} \cap \mathcal{U}$ is a $C^2$ manifold of dimension $d_{\mathcal{M}} = \dim(\mathcal{X}) + \dim(Ker(\partial_{yy}^2 g(x_0, y_0)))$. We know that $\mathcal{M} \cap \mathcal{U}$ is characterized locally by the equation $\partial_y g(x_0, y_0) = 0$, hence the tangent space $T\mathcal{M}_{(x_0, y_0)}$ of $\mathcal{M} \cap \mathcal{U}$ at point $(x_0, y_0)$ consist of the set of directions $(u, v) \in \mathcal{X} \times \mathcal{Y}$ for which $\partial_y g(x_0 + \epsilon u, y_0 + \epsilon v) = O(\epsilon^2)$. In other words $T\mathcal{M}_{(x_0, y_0)}$ is the set of vectors $(u, v) \in \mathcal{X} \times \mathcal{Y}$ of $\mathcal{M} \cap \mathcal{U}$ satisfying the equation:

$$u^\top \partial_{xy}^2 g(x_0, y_0) + v^\top \partial_{yy}^2 g(x_0, y_0) = u^\top B + v^\top A = 0.$$

Since $\mathcal{M} \cap \mathcal{U}$ is of dimension $d_{\mathcal{M}}$, the tangent space $T\mathcal{M}_{(x_0, y_0)}$ must also have dimension $d_{\mathcal{M}}$. Therefore, by the rank theorem, it must hold that the matrix $D = (B, A)$ has a rank equal to $\dim(\mathcal{X}) + \dim(\mathcal{Y}) - d_{\mathcal{M}} = \text{rank}(A)$. On the other hand, we know that $0^\top B + v^\top A = v^\top A \in Range(A)$ for any $v \in \mathcal{Y}$, so that $Range(A) \subset Range(D)$. The two subspaces having the same dimension, the inclusion implies equality ($Range(A) = Range(D)$). Henceforth, there must exist a matrix $U$ such that $B$ can be written as $B=UA$. $\qquad\square$

**Proposition 7** (**Pointwise Morse-Bott property**). *Let $g$ be a parametric Morse-Bott function. Then for any $x \in \mathcal{X}$, the function $y \mapsto g(x, y)$ is a Morse-Bott function in the following sense: For any*

$x_0$ and any critical point $y_0$ of $g(x_0, .)$, there exists an open neighborhood $\mathcal{V}$ of $y_0$ so that $C_{x,y_0} := \{y \in \mathcal{Y} | \partial_y g(x_0, y) = 0\} \cap \mathcal{V}$ is a connected sub-manifold of dimension equal to the dimension of the null space of the Hessian $\partial_{yy}^2 g(x, y_0)$.

*Proof.* Let $(x_0, y_0)$ be in $\mathcal{X} \times \mathcal{Y}$ such that $\partial_y g(x_0, y_0) = 0$. Then, since $g$ is a parameteric Morse-Bott function, there exists a neighborhood $\mathcal{U}$ of $(x_0, y_0)$ such that the augmented critical set $\mathcal{M} \cap \mathcal{U}$ is a $C^2$ manifold of dimension $d_\mathcal{M} = \dim(\mathcal{X}) + \dim(Ker(\partial_{yy}^2 g(x_0, y_0)))$. On the other hand, we know that $\mathcal{M} \cap \mathcal{U}$ is characterized locally by the equation $\partial_y g(x, y) = 0$, hence the tangent vectors $(u, v) \in \mathcal{X} \times \mathcal{Y}$ of $\mathcal{M} \cap \mathcal{U}$ at $(x_0, y_0)$ must satisfy the equation:

$$u^\top \partial_{xy}^2 g(x_0, y_0) + v^\top \partial_{yy}^2 g(x_0, y_0) = 0.$$

For simplicity, we denote by $B = \partial_{xy}^2 g(x_0, y_0)$ and $A = \partial_{yy}^2 g(x_0, y_0)$. By Proposition 6, we know that $B$ can be written in the form $B = UA$ for some matrix. Hence, the tangent space of $\mathcal{M}$ at $(x_0, y_0)$ consists in vectors $(u, v) \in \mathcal{X} \times \mathcal{Y}$ satisfying

$$\left(u^\top U + v\right)A = 0.$$

In particular, for any $u \in \mathcal{X}$, we can set $v = -u^\top U$ which ensures that $(u, v)$ is in the tangent space of $\mathcal{M}$ at $(x_0, y_0)$. Now consider the sub-manifold $\{x_0\} \times \mathcal{Y}$, its tangent space at $(x_0, y_0)$ is $\{0\} \times \mathcal{Y}$. For any element $(x, y) \in \mathcal{X} \times \mathcal{Y}$, we have the decomposition $(x, y) = (x, -x^\top U) + (0, y + x^\top U)$ where the first tuple belongs to the tangent space of $\mathcal{M}$ and the second one belongs to the tangent space of $\{x_0\} \times \mathcal{Y}$ at $(x_0, y_0)$. Hence, the tangent space of $\mathcal{X} \times \mathcal{Y}$ is generated by the both separate tangent spaces which means that both manifolds intersect transversally and that $\{x_0\} \times \mathcal{C} := (\mathcal{M} \cap \mathcal{U}) \cap (\{x_0\} \times \mathcal{Y})$ is a sub-manifold of dimension $\dim\left(Ker\left(\partial_{yy}^2 g(x_0, y_0)\right)\right)$ [29, Theorem 6.30]. For a small enough open connected neighborhood $\mathcal{V}$ of $y_0$, we can ensure that $\mathcal{C} \cap \mathcal{V}$ is a connected sub-manifold of $\mathcal{Y}$. This precisely means that $y \mapsto g(x_0, y)$ is Morse-Bott at the point $y_0$ which concludes the proof. $\qquad \square$

**Proposition 8** (**Morse functions with parameters**). *Let $g : \mathcal{X} \times \mathcal{Y}$ be a three-times continuously differentiable function such that for any $(x, y) \in \mathcal{X} \times \mathcal{Y}$ for which $\partial_y g(x, y) = 0$, the Hessian matrix $\partial_{yy}^2 g(x, y)$ is invertible. Then $g$ is a parametric Morse-Bott function.*

*Proof.* Let $(x_0, y_0) \in \mathcal{X} \times \mathcal{Y}$ be such that $y_0$ is a critical point of $g(x_0, .)$ ( i.e. $\partial_y g(x_0, y_0) = 0$). Since, by assumption, the Hessian is invertible, we can apply the implicit function theorem which guarantees the existence of a function $x \mapsto y(x)$ defined in a neighborhood $\mathcal{U}$ of $x_0$ and taking values in a neighborhood $\mathcal{V}$ of $y_0$, such that $y(x_0) = y_0$ and $y(x)$ is the unique critical point of $g(x, .)$ on $\mathcal{V}$, i.e.:

$$\partial_y g(x, y(x)) = 0, \qquad \forall x \in \mathcal{U}.$$

Moreover, $x \mapsto y(x)$ is twice continuously differentiable. This ensures that $\mathcal{M}$ the set of augmented critical points of $g$ satisfies:

$$\mathcal{M} \cap (\mathcal{U} \times \mathcal{V}) = \{(x, y(x)) \in \mathcal{X} \times \mathcal{Y} | x \in \mathcal{U}\} := \mathcal{S}.$$

We only need to show that $\mathcal{M} \cap (\mathcal{U} \times \mathcal{V})$ is a manifold of dimension $\dim(\mathcal{X})$. For this, we will apply the regular level set theorem [29, Corollary 5.14] to the function $G : (x, y) \mapsto \partial_y g(x, y)$ defined on $\mathcal{U} \times \mathcal{V}$. The pre-image of 0 by $G$ is exactly equal to $\mathcal{M} \cap (\mathcal{U} \times \mathcal{V})$. Moreover, for any $(x, y) \in \mathcal{M} \cap (\mathcal{U} \times \mathcal{V})$, we have that $dG(x, y)$ is of maximal rank since $\partial_{yy}^2 g(x, y)$ is invertible. Hence, by application of the regular level set theorem theorem to the twice continuously differentiable ($C^2$) function $G$, it follows that $\mathcal{M} \cap (\mathcal{U} \times \mathcal{V}) = G^{-1}(\{0\})$ is a $C^2$ sub-manifold of $\mathcal{X} \times \mathcal{Y}$ of dimension $\dim(ker(dG(x, y))) = \dim(\mathcal{X})$. We have shown that $\mathcal{M} \cap (\mathcal{U} \times \mathcal{V})$ is sub-manifold of dimension $\dim(\mathcal{X})$, which proves the result. $\qquad \square$

**Lemma 1.** *Let $h$ be a smooth Morse-Bott function defined on $\mathcal{Y}$. Let $\mathcal{T} : \mathcal{X} \times \mathcal{Y} \to \mathcal{Y}$ be a smooth function, such that $y \mapsto \mathcal{T}(x, y) = \tau_x(y)$ is a diffeomorphism on $\mathcal{Y}$ for any $x \in \mathcal{X}$. Then the function $g(x, y) = h(\tau_x(y))$ is a parametric Morse-Bott function.*

*Proof.* Consider the function $G : (x, y) \mapsto \partial_y g(x, y)$. We have the following equivalence

$$(x, y) \in G^{-1}(\{0\}) \iff \partial_y h(\tau_x(h)) \partial_y \tau_x(y) = 0 \iff \tau_x(y) \in \partial_y h^{-1}(\{0\}).$$

Consider the map $\mathcal{T} : (x, y) \mapsto \tau_x(y)$, then we have shown that $G^{-1}(\{0\}) = \mathcal{T}^{-1}\left(\partial_y h^{-1}(\{0\})\right)$. Let $(x_0, y_0) \in \mathcal{X} \times \mathcal{Y}$ be an augmented critical point of $g$. Set $\tilde{y} = \mathcal{T}(x_0, y_0)$ which is a critical point of $h$. Since $h$ is, by assumption, Morse-Bott at $\tilde{y}$, then there exists an open neighborhood $\tilde{\mathcal{V}}$ of $\tilde{y}$ such that $\partial_y h^{-1}(\{0\}) \cap \tilde{\mathcal{V}}$ is a sub-manifold of dimension $\dim(Ker(\partial_{yy}^2 h(\tilde{y})))$. By continuity of $\mathcal{T}$, we can always find open connected neighborhoods $\mathcal{U}$ and $\mathcal{V}$ of $(x_0, y_0)$ so that $\mathcal{V}' := \mathcal{T}(\mathcal{U} \times \mathcal{V}) \subset \tilde{\mathcal{V}}$. Moreover, since for any $x$ $\mathcal{T}(x, .)$ is a diffeomorphism, it must be that $\mathcal{V}'$ is an open set. Therefore, $\mathcal{S} := \partial_y h^{-1}(\{0\}) \cap \mathcal{V}'$ must be a sub-manifold as of dimension $\dim(Ker(\partial_{yy}^2 h(\tilde{y})))$. It remains to show that $\mathcal{T}^{-1}(\mathcal{S})$ is a sub-manifold. To see this, it suffice to note that the differential of $\mathcal{T}$ is surjective which ensures that $\mathcal{T}$ is transverse to $\mathcal{S}$ and that $\mathcal{T}^{-1}(\mathcal{S})$ is a sub-manifold [29, Theorem 6.30]. Moreover, the dimension of such manifold is equal to $\dim(\mathcal{X}) + \dim(ker(\partial_{yy}^2 h(\tilde{y}))) = \dim(\mathcal{X}) + \dim(ker(\partial_{yy}^2 g(x_0, y_0)))$.

$\square$

### A.3 Proof of the Morse-Bott Lemma with Parameters

In this section, we provide a proof of the Morse-Bott lemma with parameters introduced in Theorem 1. We then introduces two results in Proposition 9 and Corollary 1 which are consequences of Theorem 1. Proposition 9 shows that near an augmented critical point $(x_0, y_0)$, the Hessian matrices of nearby augmented critical points are all similar. This result illustrates that the geometry near a critical point is preserved when the parameter $x$ is perturbed. Proposition 9 will be used later in Proposition 16 of Appendix C to show that the pseudo-inverse of the Hessian matrices of critical points near a local minimum are uniformly bounded. Finally, Corollary 1 shows that near any augmented critical point $(x_0, y_0)$ the function $y \mapsto g(x, y)$ can be expressed as a slight deformation of $y \mapsto g(x_0, y)$. This result, along with the stability result in Appendix B.2 of the gradient flow to deformations will be key to prove the continuity of the selection map $x \mapsto \phi(x, y)$ near local minima.

*Proof of Theorem 1.* Let $x_0 \in \mathcal{X}$ and $y_0$ be a critical point of $g(x_0, .)$. Denote by $\mathcal{K}$ the null space of the Hessian $A_0 = \partial_{yy}^2 g(x_0, y_0)$ and by $\mathcal{K}^\perp$ its orthogonal complement in $\mathcal{Y}$. The function $g(x_0, .)$ is a Morse-Bott function by Proposition 7, therefore by the Morse-Bott lemma [16, Theorem 2.10], there exists three open neighborhoods $\mathcal{O}$, $\mathcal{O}^\perp$ and $\mathcal{V}$ of $0 \in \mathcal{K}$, $0 \in \mathcal{K}^\perp$ and $y_0 \in \mathcal{Y}$ and a diffeomorphism $s : \mathcal{O} \times \mathcal{O}^\perp \to \mathcal{V}$ s.t. $s(0, 0) = y_0$ and for any $r, w \in \mathcal{O} \times \mathcal{O}^\perp$ it holds that:

$$g(x_0, s(r, w)) = g(x_0, y_0) + \frac{1}{2} w^\top J_0 w, \forall r, w \in \mathcal{O} \times \mathcal{O}^\perp.$$

where $J_0$ is an invertible diagonal matrix whose diagonal elements are equal to the sign of the non-zero eigenvalues of the Hessian $\partial_{yy}^2 g(x_0, y_0)$. By convention $J_0 = 0$ in case the Hessian $\partial_{yy}^2 g(x_0, y_0) = 0$. Since, the function $h(x, r, w) := g(x, s(r, w))$ is such that $\partial_w h(x_0, 0, 0) = 0$ and the partial Hessian $\partial_{ww}^2 h(x_0, 0, 0) = J_0$ is invertible, we are in position to apply the Morse lemma with parameters [16, Theorem 4]. The lemma ensures that $\mathcal{O}$ and $\mathcal{O}^\perp$ can be chosen small enough so that there exits open neighborhoods $\mathcal{B}$ and $\mathcal{O}_1^\perp$ of $x_0 \in \mathcal{X}$ and $0 \in \mathcal{K}^\perp$ and a diffeomorphism $\tau$ from $\mathcal{B} \times \mathcal{O} \times \mathcal{O}_1^\perp$ to $\mathcal{B} \times \mathcal{O} \times \mathcal{O}^\perp$ such that $\tau(x_0, 0, 0) = (x_0, 0, 0)$ and decomposing $h$ locally into a quadratic component and a singular one. More precisely, for any $(x, r, w) \in \mathcal{B} \times \mathcal{O} \times \mathcal{O}_1^\perp$, the map $r$ satisfies $\tau(x, r, w) = (x, r, w')$ for some $w' \in \mathcal{O}^\perp$ and the following equation holds:

$$h(\tau(x, r, w)) = h(\tau(x, r, 0)) + \frac{1}{2} w^\top J_0 w. \tag{9}$$

It remains to show that $\xi \mapsto h(\tau(x, r, 0))$ is in fact constant for $(x, r)$ in an open neighborhood of $(x_0, 0) \in \mathcal{X} \times \mathcal{O}$. To this end, define the sets $A$, $B$ and $C$ as follows:

$$
\begin{aligned}
A &:= \{(x, y) \in \mathcal{B} \times \mathcal{V} \quad | \quad \partial_y g(x, y) = 0\}, \\
B &:= \{(x, r, w) \quad | \quad (x, r) \in \mathcal{B} \times \mathcal{O}, \quad \partial_r h(\tau(x, r, w)) = 0\}, \\
C &:= \{(x, r, 0) \quad | \quad (x, r) \in \mathcal{B} \times \mathcal{O}, \quad \partial_r h(\tau(x, r, 0)) = 0\}.
\end{aligned}
$$

Then by (9), it holds that $B = C$. Moreover, $A$ and $B$ are homeomorphic. Indeed to see this, we introduce the notation $\tilde{s}(x, r, w) := (x, s(r, w))$ which defines a diffeomorphism from $\mathcal{B} \times \mathcal{O} \times \mathcal{O}^\perp$ to $\mathcal{B} \times \mathcal{V}$. Hence, $g \circ \tilde{s} \circ \tau = h \circ \tau$. This ensures $\tilde{s} \circ \tau(B) = A$, which means precisely that

$A$ and $B$ are homeomorphic since $\tilde{s} \circ \tau$ is a homeomorphism. Moreover, by definition of $g$ as a parametric Morse-Bott function, we also know that $A$ is a sub-manifold of $\mathcal{X} \times \mathcal{Y}$ of dimension $\dim(\mathcal{X}) + \dim\left(Ker\left(\partial_{yy}^2 g(x_0, y_0)\right)\right)$ provided the neighborhoods $\mathcal{B}$ and $\mathcal{V}$ are small enough. Hence, we can deduce that $B$ and $C$ must also be sub-manifolds of the same dimension. In particular, $C$ is a sub-manifold of $\mathcal{B} \times \mathcal{O} \times \{0\}$ which is of dimension $\dim(\mathcal{X}) + \dim\left(Ker\left(\partial_{yy}^2 g(x_0, y_0)\right)\right)$. Therefore, $C$ is an open sub-manifold of $\mathcal{B} \times \mathcal{O} \times \{0\}$. Hence, since $(x_0, 0, 0) \in C$, there must exists an open connected neighborhood $\mathcal{B}_1 \times \mathcal{O}_1 \times \{0\}$ of $(x_0, 0, 0)$ in $\mathcal{B} \times \mathcal{O} \times \{0\}$ that is contained in $C$. Hence, we deduce that for any $(x, r) \in \mathcal{B}_1 \times \mathcal{O}_1$, the function $h$ satisfies $\partial_r h(\tau(x, r, 0)) = 0$ so that $h(\tau(x, r, 0)) = h(\tau(x, 0, 0))$ on such neighborhood. Finally, we have shown that there exits

$$g \circ \tilde{s} \circ \tau(x, r, w) = g \circ \tilde{s} \circ \tau(x, 0, 0) + \frac{1}{2} w^\top J_0 w.$$

We conclude the proof by setting $\psi(x, r, w) = \tilde{s} \circ \tau(x, r, w)$ which is the desired diffeomorphism. $\qquad \square$

**Proposition 9.** *Let $g$ be a real-valued function such that Assumption 1 holds. Consider an augmented critical point $(x_0, y_0) \in \mathcal{M}$, with $\mathcal{M}$ defined in (8). Then there exists a neighborhood $\mathcal{V}$ of $(x_0, y_0)$ and a continuous map $(x, y) \mapsto P(x, y)$ defined on $\mathcal{V}$ with values in $\mathbb{R}^{d \times d}$ such that:*

- *$P(x, y)$ is invertible for any $(x, y) \in \mathcal{V}$ with singular values contained in an interval $[\sigma_{\min}, \sigma_{\max}]$ for some positive constants $\sigma_{\min}$ and $\sigma_{\max}$.*

- *For any augmented critical point $(x, y) \in \mathcal{V}$, the Hessian of $g$ is given by:*

$$\partial_{yy}^2 g(x, y) = P(x, y)^\top \partial_{yy}^2 g(x_0, y_0) P(x, y).$$

*Proof.* Denote by $\mathcal{K}$ the null space of the Hessian $A_0 = \partial_{yy}^2 g(x_0, y_0)$ and by $\mathcal{K}^\perp$ its orthogonal complement in $\mathcal{Y}$. Let $J_0$ be a diagonal matrix with diagonal elements given by the sign of the non-zero eigenvalues of $A_0$. Since $g$ satisfies Assumption 1, we apply Theorem 1 which ensures the existence of a diffeomorphism $\psi$ defined on an open neighborhood $\mathcal{U}$ of $(x_0, 0, 0) \in \mathcal{X} \times \mathcal{K} \times \mathcal{K}^\perp$ with values in an open neighborhood $\mathcal{V}$ of $(x_0, y_0)$ in $\mathcal{X} \times \mathcal{Y}$, s.t. $\psi(x_0, 0, 0) = (x_0, y_0)$ and for all $(x, r, w) \in \mathcal{U}$, $\psi$ satisfies $\psi(x, r, w) = (x, y)$ and

$$g(\psi(x, r, w)) = g(\psi(x, 0, 0)) + \frac{1}{2} w^\top J_0 w, \tag{10}$$

$$= g(\psi(x, 0, 0)) + \frac{1}{2}(r^\top w^\top) \tilde{J}_0 \begin{pmatrix} r \\ w \end{pmatrix},$$

where we defined $\tilde{J}_0$ to be the matrix of dimension $d \times d$ given by:

$$\tilde{J}_0 = \begin{pmatrix} 0 & 0 \\ 0 & J_0 \end{pmatrix}.$$

Since $\psi$ is a diffeomorphism satisfying $\psi(x, r, w) = (x, y)$, we can equivalently write (10) as:

$$g(x, y) = g(\psi(x, 0, 0)) + \frac{1}{2} \psi_{2,3}^{-1}(x, y)^\top \tilde{J}_0 \psi_{2,3}^{-1}(x, y), \qquad \forall (x, y) \in \mathcal{V}, \tag{11}$$

where $\psi_{2,3}^{-1}(x, y)$ are last two components of $\psi^{-1}(x, y)$ (i.e. $\psi^{-1}(x, y) = (x, \psi_{2,3}^1(x, y))$). By differentiating (11) w.r.t. $y$ we obtain:

$$\partial_y g(x, y) = \partial_y \psi_{2,3}^{-1}(x, y) \tilde{J}_0 \psi_{2,3}^{-1}(x, y). \tag{12}$$

$\partial_y \psi_{2,3}^{-1}(x, y)$ must be invertible since $(\partial_x \psi^{-1}, \partial_y \psi^{-1})$ is invertible and of the form:

$$\begin{pmatrix} \partial_x \psi^{-1} \\ \partial_y \psi^{-1} \end{pmatrix} = \begin{pmatrix} I & \partial_x \psi_{2,3}^{-1} \\ 0 & \partial_y \psi_{2,3}^{-1} \end{pmatrix}.$$

Therefore, if $y$ is a critical point of $g(x, .)$, then (12) implies that $\tilde{J}_0 \psi_{2,3}^{-1}(x, y) = 0$. Let $(x, y)$ be an augmented critical point of $g$, $\epsilon > 0$ and $u$ be vector in $\mathcal{Y}$, then the following holds:

$$\frac{1}{\epsilon} \partial_y g(x, y + \epsilon u) = \left(\partial_y \psi_{2,3}^{-1}(x, y + \epsilon u)\right)^\top \tilde{J}_0 \left(\frac{1}{\epsilon} \psi_3^{-1}(x, y + \epsilon u) - \psi_{2,3}^{-1}(x, y)\right).$$

Hence, by taking the limit when $\epsilon$ approaches 0, it follows that:

$$\partial_{yy}^2 g(x,y) = \left(\partial_y \psi_{2,3}^{-1}(x,y)\right)^\top \tilde{J}_0 \partial_y \psi_{2,3}^{-1}(x,y).$$

Define $P_0 := \partial_y \psi_{2,3}^{-1}(x_0, y_0) \partial_y \psi_{2,3}^{-1}(x_0, y_0)^\top$ which is invertible. Then, we can write:

$$\begin{aligned}
\partial_{yy}^2 g(x,y) &= \partial_y \psi_{2,3}^{-1}(x,y)^\top P_0^{-1} P_0 \tilde{J}_0 P_0 P_0^{-1} \partial_y \psi_{2,3}^{-1}(x,y) \\
&= \partial_y \psi_{2,3}^{-1}(x,y)^\top P_0^{-1} \partial_y \psi_{2,3}^{-1}(x_0, y_0) A_0 \partial_y \psi_{2,3}^{-1}(x_0, y_0)^\top P_0^{-1} \partial_y \psi_{2,3}^{-1}(x,y) \\
&= P(x,y)^\top \partial_{yy}^2 g(x_0, y_0) P(x,y),
\end{aligned}$$

where we defined $P(x,y) = \partial_y \psi_{2,3}^{-1}(x_0, y_0)^\top P_0^{-1} \partial_y \psi_{2,3}^{-1}(x,y)$. The matrix $P(x,y)$ is invertible for any $(x,y) \in \mathcal{V}$ and the map $(x,y) \mapsto P(x,y)$ is continuous. Hence, by considering compact neighborhood of $(x_0, y_0)$ contained in $\mathcal{V}$, we can ensure that the singular values of $P(x,y)$ are contained in an interval $[\sigma_{\min}, \sigma_{\max}]$ where $\sigma_{\min}$ and $\sigma_{\max}$ are positive numbers. Further considering the restriction of such map on an open neighborhood $\mathcal{V}' \subset K$ of $(x_0, y_0)$ yields the desired result. $\qquad \square$

**Corollary 1.** *Let $g$ be a real-valued function such that Assumption 1 holds. Consider an augmented critical point $(x_0, y_0) \in \mathcal{M}$, with $\mathcal{M}$ defined in (8). Then, there exists a open neighborhoods $\mathcal{B}$ and $\mathcal{V}$ of $x_0$ and $y_0$ in $\mathcal{X}$ and $\mathcal{Y}$ and a continuously differentiable map $\tau$ from $\mathcal{B} \times \mathcal{V}$ to $\mathcal{V}$ such that:*

- *For any $x \in \mathcal{B}$, the map $\tau_x : y \mapsto \tau(x,y)$ is a diffeomorphism from $\mathcal{V}$ to itself satisfying $\tau_{x_0}(y) = y$ for any $y \in \mathcal{V}$. Moreover, $(x,y) \mapsto \tau_x^{-1}(y)$ is continuous.*

- *For any $(x,y) \in \mathcal{B} \times \mathcal{V}$, the function $g$ satisfies $g(x,y) = g(x_0, \tau(x,y)) + C(x)$, where $x \mapsto C(x)$ is a function independent of $y$.*

- *There exists positive numbers $\ell$ and $L$ s.t for any $(x,y) \in \mathcal{B} \times \mathcal{V}$:*

$$\ell^2 I \le \partial_y \tau(x,y)^\top \partial_y \tau(x,y) \le (L')^2 I. \tag{13}$$

*Proof.* We use the notations of Theorem 1 where $\mathcal{K}$ is the null subspace of the Hessian $\partial_{yy}^2 g(x_0, y_0)$ and $\mathcal{K}^\perp$ its orthogonal complement in $\mathcal{Y}$. By Theorem 1 $g$ satisfies:

$$g(\psi(x,r,w)) = g(\psi(x,0,0)) + \frac{1}{2} \bar{y}^\top J_0 \bar{y},$$

with $\psi$ and $J_0$ being the diffeomorphism and matrix defined in Theorem 1. Recall that $\psi$ is defined on an open neighborhood $\mathcal{B} \times \mathcal{O} \times \mathcal{O}^\perp$ of $(x_0, 0, 0) \in \mathcal{X} \times \mathcal{K} \times \mathcal{K}^\perp$ and whose image by $\psi$ is an open neighborhood $\mathcal{B} \times \mathcal{V}$ of $(x_0, y_0)$. Hence, we can write:

$$g(\psi(x,r,w)) = C(x) + g(\psi(x_0, r, w)),$$

with $C(x) := g(\psi(x,0,0)) - g(\psi(x_0, 0, 0))$. We also know that $\psi$ preserves $x$, meaning that $\psi(x,r,w) = (x,y)$. Hence, we can define $(x,r,w) \mapsto \tilde{\tau}_x(r,w) \in \mathcal{Y}$, s.t. $\psi(x,r,w) = (x, \tilde{\tau}_x(r,w))$. For any $x \in \mathcal{B}$, $(r,w) \mapsto \tilde{\tau}_x(r,w)$ defines a diffeomorphism from $\mathcal{O} \times \mathcal{O}^\perp$ onto its image. Moreover, its image must be equal to $\mathcal{V}$. Indeed, since $\psi(\mathcal{B} \times \mathcal{O} \times \mathcal{O}^\perp) = \mathcal{B} \times \mathcal{V}$, it follows that for any $(x,y) \in \mathcal{B} \times \mathcal{V}$, there exists $(r,w) \in \mathcal{O} \times \mathcal{O}^\perp$ such that $\psi(x,r,w) = (x, \tilde{\tau}_x(r,w)) = (x,y)$. In particular, if $(x,y) \in \mathcal{B} \times \mathcal{V}$ and $(r,w) = \tilde{\tau}_x^{-1}(y)$, we can write $\psi(x_0, r, w) = (x_0, \tilde{\tau}_{x_0}(r,w)) = (x_0, \tilde{\tau}_{x_0} \tilde{\tau}_x^{-1}(y))$. Therefore, the following expression holds for any $(x,y) \in \mathcal{B} \times \mathcal{V}$:

$$g(x,y) = C(x) + g(x_0, \tau(x,y)),$$

where we defined $\tau(x,y) = \tilde{\tau}_{x_0} \circ \tilde{\tau}_x^{-1}(y)$. For any $x \in \mathcal{B}$, the map $\tau_x : y \mapsto \tau(x,y)$ is a diffeomorphism satisfying $\tau(x_0, y) = y$. Moreover, $(x,r,w) \mapsto \tilde{\tau}_x(r,w)$ and $(x,y) \mapsto \tilde{\tau}_x^{-1}(y)$ are continuously differentiable since $\psi$ is a diffeomorphism. As a result, $\tau$ is continuously differentiable as well and $(x,y) \mapsto \tau_x^{-1}(y)$ is continuously differentiable. Finally, since $\partial_y \tau(x,y)$ is jointly continuous in $x$ and $y$ and $\partial_y \psi_x(y)$ is invertible, then, provided that $\mathcal{B}$ and $\mathcal{V}$ are small enough, there must exist two positive numbers $\ell$ and $L'$ such that for any $(x,y) \in \mathcal{B} \times \mathcal{V}$:

$$\ell^2 I \le \partial_y \psi(y)^\top \partial_y \psi(y) \le (L')^2 I.$$

$\qquad \square$

*Proof of Proposition 1* . Recall $\mathcal{M} = \{(x, y) \in \mathcal{X} \times \mathcal{Y} | \partial_y g(x, y) = 0\}$ the set of augmented critical points of $g$ and let $(x_0, y_0)$ be in $\mathcal{M}$. First, since Assumption 1 holds, we know by Proposition 7 that $g(x_0, .)$ is a Morse-Bott function. Hence, by [16, Theorem 1], it follows that $g(x_0, .)$ satisfies a Łojasiewicz inequality near $y_0$. In other words, there exists a neighborhood $\mathcal{V}$ of $y_0$ and a positive constant $\mu' > 0$ such that:

$$\mu'|g(x_0, y) - g(x_0, y_0)| \leq \frac{1}{2}\|\partial_y g(x_0, y)\|^2, \qquad \forall y \in \mathcal{V}.$$

By Corollary 1, there exists a continuous function $\tau$ defined on an open neighborhood $\mathcal{B} \times \mathcal{V}$ of $(x_0, y_0)$ whose image is $\mathcal{V}$ and for which $g(x, y) = g(x_0, \tau(x, y)) + C(x)$ for any $(x, y) \in \mathcal{B} \times \mathcal{V}$, where $C(x)$ is a function of $x$ independent of $y$. Moreover, for any $x \in \mathcal{B}$, $y \mapsto \tau(x, y)$ is a diffeomorphism from $\mathcal{V}$ to itself whose inverse is written as $\tau^{-1}(x, y)$ by an abuse of notion. In particular, for $y = \tau^{-1}(x, y_0)$ we set $G(x) := g(x, \tau^{-1}(x, y_0)) = g(x_0, y_0) + C(x)$. Note that $\tau^{-1}(x, y_0)$ is critical point of $g(x, .)$ since $\partial_y g(x, \tau^{-1}(x, y_0))\partial_y \tau^{-1}(x, y_0) = \partial_y g(x_0, y_0) = 0$ and $\partial_y \tau^{-1}(x, y_0)$ is invertible. Hence, the following holds for any $(x, y) \in \mathcal{B} \times \mathcal{V}$.

$$\mu'|g(x, y) - G(x)| = \mu'|g(x_0, \tau(x, y)) - g(x_0, y_0)| \leq \frac{1}{2}\|\partial_y g(x_0, \tau(x, y))\|^2. \tag{14}$$

Moreover, by construction of $\tau$, we know that $\partial_y \tau(x, y)$ satisfies (13) for any $(x, y) \in \mathcal{B} \times \mathcal{V}$. Therefore, we deduce that:

$$\|\partial_y g(x, y)\|^2 = \|\partial_y g(x_0, \tau(x, y))\partial_y \tau(x, y)\|^2 \geq \ell^2 \|\partial_y g(x_0, \tau(x, y))\|^2,$$

Finally, combining the above inequality with (14), we get that, for any $(x, y) \in \mathcal{B} \times \mathcal{V}$:

$$\ell^2 \mu'|g(x, y) - G(x)| \leq \frac{1}{2}\|\partial_y g(x, y)\|^2, \qquad \forall (x, y) \in \mathcal{U}.$$

The result follows by setting $\mu = \ell^2 \mu' > 0$ and $\mathcal{U} = \mathcal{B} \times \mathcal{V}$. $\qquad\square$

# B  Asymptotic Properties of Gradient Flows

## B.1  Convergence of the gradient flow.

Recall that the gradient flow $\phi_t(x, y)$ satisfies the differential equation

$$\frac{d\phi_t(x, y)}{dt} = -\partial_y g(x, \phi_t(x, y)), \quad \phi_0(x, y) = y.$$

The next proposition shows that the gradient flow $\phi_t(x, y)$ converges towards a well-defined selection map $\phi(x, y)$.

**Proposition 10 (Convergence of $\phi_t$.).** *Let $x, y$ be in $\mathcal{X} \times \mathcal{Y}$. Under Assumptions 1 to 3, $(t, x, y) \mapsto \phi_t(x, y)$ is continuous and for any $(x, y) \in \mathcal{X} \times \mathcal{Y}$, $\phi_t(x, y)$ converges towards a unique critical point $\phi(x, y)$ of $y \mapsto g(x, y)$ as $t$ goes to $+\infty$.*

*Proof.* First, Assumption 2 ensures that the gradient flow $\phi_t(x, y)$ is uniquely defined at all times $t$ [12]. $\phi_t(x, y)$ is jointly continuous in $(t, x, y)$ by Cauchy-Lipschitz theorem. Moreover, $t \mapsto \phi_t(x, y)$ remains bounded thanks to Assumption 3. Otherwise, there exists a subsequence $\phi_{t_n}(x, y)$ such that $g(x, \phi_{t_n}(x, y))$ diverges to $+\infty$. This contradicts the fact that $g(x, \phi_{t_n}(x, y))$ is decreasing since $\phi_t(x, y)$ is a gradient flow of $g$. Hence, we deduce that $\phi_t(x, y)$ must have at least one accumulation point $y^\star$. Moreover, $y^\star$ must be a critical point of $g(x, .)$. To see this, note that $g(x, \phi_t(x, y))$ is a decreasing function in time and is lower-bounded. Hence, it admits a finite limit $l$. Moreover, by differentiating $g(x, \phi_t(x, y))$ is time, it follows that:

$$\frac{\mathrm{d}}{\mathrm{d}t} g(x, \phi_t(x, y)) = -\|\partial_y g(x, \phi_t(x, y))\|^2$$

This implies that $\int_0^{+\infty} \|\partial_y g(x, \phi_s(x, y))\|^2 \,\mathrm{d}s = g(x, y) - l$ is finite. Since, $g$ is $L$-smooth by Assumption 2, this is only possible if $\partial_y g(x, \phi_s(x, y))$ converges to 0. In particular, by continuity of $\partial_y g(x, y)$, it follows that $\partial_y g(x, y^\star) = 0$. We only need to show that $y^\star$ is the unique accumulation

point of $\phi_t(x, y)$. To show this, we apply Proposition 1, which implies, in particular, that $g$ satisfies a Łojasiewicz inequality in a neighborhood $\mathcal{V}$ of $y^\star$:

$$\mu|g(x, y) - G(x)| \leq \|\partial_y g(x, y)\|^2, \forall y \in \mathcal{V}.$$

We can therefore apply [40, Theorem 2.7] which ensure that $y^\star$ is the unique accumulation point of $\phi_t(x, y)$ and that $\phi_t(x, y)$ converges towards $y^\star$. We can therefore defined the map $\phi(x, y) = \lim_{t\to\infty} \phi_t(x, y)$ which constitues a selection.

$\square$

## B.2   Stability of the Gradient Flow Near Local Minima

In this section, we provide a general result establishing the stability of gradient flows to perturbations. This result shows that deforming a gradient flow by a family of diffeomorphisms yields trajectories that are not too far from the unperturbed flow. We will use this result later in Appendix B.3 in conjunction with the formulation of $y \mapsto g(x, y)$ as a perturbation of $y \mapsto g(x_0, y)$ provided in Corollary 1 to prove that the gradient flow $\phi_t(x, y)$ remain stable as the parameter $x$ varies.

**Proposition 11** (**Stability near local minima**). *Let $h$ be a real valued differentiable function defined on $\mathcal{Y}$ and $y_0$ be a local minimizer of $h$. We assume that $h$ satisfies the Łojasiewicz inequality near $y_0$, meaning that there exists $\mu > 0$ and $R > 0$ s.t.:*

$$\mu(h(y) - h(y_0)) \leq \frac{1}{2}\|\partial_y h(y)\|^2, \qquad \forall y \in B(y_0, R). \tag{15}$$

*Let $\mathcal{V}$ be an open neighborhood of $y_0$, $R' > 0$ such that $B(y_0, 2R') \subset \mathcal{V}$ and $\mathcal{P}$ a family of diffeomorphisms defined from $\mathcal{V}$ to itself and satisfying:*

1. *For any $\tau \in \mathcal{P}$, the pre-image $y_\psi := \psi^{-1}(y_0)$ of $y_0$ by $\psi$ belongs to $B(y_0, R')$.*

2. *There exists positive numbers $\ell$ and $L'$ s.t. for any $\tau \in \mathcal{P}$ and any $y \in \mathcal{V}$:*

$$\ell^2 I \leq \partial_y \tau(y)^\top \partial_y \tau(y) \leq (L')^2 I. \tag{16}$$

*For some $\tau \in \mathcal{P}$, consider a maximal solution $(z_t)$ of the following ODE:*

$$\dot{z}_t = -\partial_y h(\tau(z_t))\partial_z \tau(z_t), \qquad z_0 \in B(y_\tau, R'). \tag{17}$$

*Then, there exists $0 < C \leq R'$, such that for any $0 < \epsilon \leq C$, there exists $0 < \eta \leq \frac{\epsilon}{2}$ with the following property:*

*For any $\tau \in \mathcal{P}$ and any $z_0$ s.t. $\|z_0 - y_\tau\| \leq \eta$:*

1. *The solution $z_t$ to (17) is well-defined at all times $t \geq 0$.*

2. *For all $t \geq 0$, it holds that $\|z_t - y_\tau\| \leq \epsilon$.*

*Proof.* The proof is inspired from the the abstract stability result in [31]. We know that $y_0$ is a local minimizer of $h$, therefore there exists $R" > 0$ such that for any $y$ satisfying $\|y - y_0\| \leq R"$, it holds that $\mathcal{L}(y) := h(y) - h(y_0) \geq 0$. Moreover, by (15), we also have that:

$$2\mu\mathcal{L}(y) \leq \|\partial_y \mathcal{L}(y)\|^2, \qquad \forall y \in B(y_0, R). \tag{18}$$

Take $\epsilon < \frac{1}{L'}\min(R, R", L'R') := C$. To simplify subsequent calculations, we will choose $y$ close enough to $y_0$ so that $2\ell^{-1}\sqrt{\frac{2}{\mu}}\mathcal{L}(y)^{\frac{1}{2}} \leq \epsilon$, where $\mu$ is the positive constant appearing in (20) and $\ell$ is the positive constant in (16). This is possible by continuity of $\mathcal{L}$ which that there exists $0 < \eta \leq \frac{\epsilon}{2}$ for which any $y \in B(y_0, L'\eta)$ satisfies:

$$\mathcal{L}(y)^{\frac{1}{2}} \leq \frac{1}{2}\ell\sqrt{\frac{\mu}{2}}\epsilon. \tag{19}$$

Consider now $\tau \in \mathcal{P}$. Equation (16) implies that $\tau$ is $L'$-Lipschitz on $B(y_0, 2R')$. Moreover, for any $z$ in $B(y_\tau, \eta)$, it holds that $z \in B(y_0, 2R')$ since $\eta \leq R'$ and $y_\tau \in B(y_0, R')$ by definition of $y_\tau$. Therefore, we can write the following inequality:

$$\|\tau(z) - y_0\| = \|\tau(z) - \tau(y_\tau)\| \leq L'\|z - y_\tau\| \leq L'\eta,$$

We have shown that $\tau(z) \in B(y_0, L'\eta)$ for any $z \in B(y_\tau, \eta)$, so that (19) holds for $\tau(z)$:

$$\mathcal{L}(\tau(z))^{\frac{1}{2}} \leq \frac{1}{2}\ell\sqrt{\frac{\mu}{2}}\epsilon, \qquad \forall z \in B(y_\tau, \eta),$$

Additionally, by (20) and using that $\ell^2\|\partial_y\mathcal{L}(\tau(z))\|^2 \leq \|\nabla\mathcal{L} \circ \tau(z)\|^2$ by (16), it holds for any $\epsilon < C$ that:

$$0 \leq 2\mu\mathcal{L}(\tau(z)) \leq \|\partial_y\mathcal{L}(\tau(z))\|^2 \leq \ell^{-2}\|\nabla\mathcal{L} \circ \tau(z)\|^2, \qquad \forall z \in B(y_\tau, \epsilon), \qquad (20)$$

From now on, we fix $\tau$, and consider $z_t$ to the ODE (17) with initial condition $z_0 \in B(y_\tau, \eta)$. Define $\mathcal{T} = \{t \in \mathbb{R}_+|.\|z_s - y_\tau\| < C \quad \forall s \in [0, t)\}$ which is not empty by construction since $\|z_0 - y_\tau\| < C$ $s \mapsto z_s$ is continuous. Hence, $t_1 := \sup \mathcal{T}$ is positive. We will show that $t_1 = +\infty$. We will also consider the time until which $\mathcal{L}(\tau(z_t))$ remains positive: $t^+ := \sup\{t \in \mathbb{R}_+|\mathcal{L}(\tau(z_s)) > 0 \forall s \in [0, t)\}$. We may assume that $\mathcal{L}(\tau(z_0)) > 0$ so that $t^+ > 0$ by continuity of the solution $z_t$. The case where $\mathcal{L}(\tau(z_0)) = 0$ will be treated separately. Denote by $t_1^+ := \min(t_1, t^+)$ so that, for any $t \in [0, t_1^+)$ the following holds:

$$-\frac{d\mathcal{L}(\tau(z_t))^{\frac{1}{2}}}{dt} = \frac{1}{2}\mathcal{L}(\tau(z_t))^{-\frac{1}{2}}\|\nabla\mathcal{L} \circ \tau(z_t)\|^2 \geq \ell\sqrt{\frac{\mu}{2}}\|\nabla\mathcal{L} \circ (\tau(z_t))\|,$$

where the first equality follows by differentiating $z_t$ in time and using the ODE equation (17), while the last inequality uses the inequality (20) which holds since $\|z_t - y_\tau\| < C$. Integrating between 0 and $t \in [0, t_1^+)$, we get:

$$\mathcal{L}(\tau(z_0))^{\frac{1}{2}} - \mathcal{L}(\tau(z_t))^{\frac{1}{2}} \geq \ell\sqrt{\frac{\mu}{2}}\int_0^t \|\nabla\mathcal{L} \circ \tau(z_s)\| \, ds.$$

Since $\|z_0 - y_\tau\| \leq \eta$ and using (19), it holds that $\mathcal{L}(z_0)^{\frac{1}{2}} \leq \frac{\ell}{2}\sqrt{\frac{\mu}{2}}\epsilon$. We can therefore deduce that $\int_0^t \|\nabla\mathcal{L} \circ \tau(z_s)\| \, ds \leq \frac{\epsilon}{2}$. This allows to write for all $t \in [0, t_1^+)$

$$\|z_t - y_\tau\| \leq \|z_t - z_0\| + \|z_0 - y_\tau\|, \qquad (21)$$
$$\leq \int_0^t \|\nabla\mathcal{L} \circ \tau(z_s)\| \, ds + \eta \leq \epsilon.$$

We distinguish two cases depending on whether $t^+ < t_1$ or $t_1 \leq t^+$.

**Case 1:** $t^+ < t_1$ **or**. In this case we have $t_1^+ = t^+ < +\infty$. This case also accounts for when $\mathcal{L} \circ \tau(z_0)=0$ which implies that $t^+ = 0 < t_1$. If $t^+=0$, then $\|z_{t^+} - y_\tau\| \leq \epsilon$ by construction. Otherwise, we still have that $\|z_{t^+} - y_\tau\| \leq \epsilon$ by (21) and the continuity of $z_t$ at $t^+$. Moreover, by definition of $t^+$, it must also hold that $\mathcal{L} \circ \tau(z_{t^+}) = 0$. We only need to show that $\nabla\mathcal{L} \circ \tau(z_{t^+}) = 0$. By contradiction, if $\nabla\mathcal{L}\circ\tau(z_{t^+}) \neq 0$, then we would have $\mathcal{L}\circ\tau(z_{t^++s}) < 0$ for $s > 0$ small enough. However, since $t^+ < t_1$, then $t^+ + s < t_1$ for $s$ small enough, so that $\|z_{t^++s} - y_\tau\| < C$. The latter means that $\mathcal{L} \circ \tau(z_{t^++s}) \geq 0$ since $y_0$ is a local minimizer of $\mathcal{L}$. This contradicts $\mathcal{L} \circ (z_{t^++s}) < 0$. Therefore $\nabla\mathcal{L}\circ\tau(z_{t^+})=0$ which implies that $\tau(z_{t^+})$ is a critical point of $y \mapsto \mathcal{L}(y)$ so that $z_t = z_{t^+}$ for any $t \geq t^+$. This directly means that $\|z_t - y_\tau\| \leq \epsilon$ for any $t \geq 0$, hence $t_1 = +\infty$.

**Case 2:** $t^+ \geq t_1$. In this case, $t_1^+ = t_1$. If by contradiction we had $t_1 < +\infty$, then we would directly get $\|z_{t_1} - y_\tau\| \leq \epsilon$ by continuity of $t$ at $t_1$ and maximality of the solution $z_t$. However, by definition of $t_1$, we also have $\|z_{t_1} - y_\tau\| = C$. This contradicts the condition $\epsilon < C$ and therefore means that $t_1 = +\infty$. Hence, it holds that $\|z_t - y_\tau\| \leq \epsilon$ for any $t \geq 0$ and that the solution $z_t$ is well-defined at all times.

$\square$

## B.3 Continuity of the Flow Selection

Proposition 12 shows that $x \mapsto \phi(x, y)$ is continuous at $x_0$ whenever $\phi(x_0, y)$ is a local minimum of $g(x_0, .)$. Proposition 13 shows that, near $x_0$, $\phi(x, y)$ are local minima as well provided $\phi(x_0, y)$ is a local minimum of $g(x_0, .)$.

**Proposition 12 (Continuity near local minima).** *Let $x_0 \in \mathcal{X}$ and $y \in \mathcal{Y}$. Let $g$ be such that Assumptions 1 to 3 hold. Assume that $y_0 = \phi(x_0, y)$ is a local minimizer of $y \mapsto g(x_0, y)$. Then, for any $\epsilon > 0$ small enough, there exists $T > 0$ and $\eta > 0$, s.t.:*

$$\|\phi_t(x, y) - \phi(x_0, y)\| \le \epsilon, \qquad \forall t \ge T, \quad \forall x \in B(x_0, \eta).$$

*In particular, $x \mapsto \phi(x, y)$ is continuous at $\phi(x_0, y)$.*

*Proof.* We will apply Proposition 11 to the function $h(y) = g(x_0, y)$ and the well-chosen family $\mathcal{P}$ of local diffeomorphisms on $\mathcal{Y}$. By application of Corollary 1, there exists a open neighborhoods $\mathcal{B}$ and $\mathcal{V}$ of $x_0$ and $y_0$ in $\mathcal{X}$ and $\mathcal{Y}$ and a continuously differentiable map $\tau$ from $\mathcal{B} \times \mathcal{V}$ to $\mathcal{V}$ such that $y \mapsto \tau(x, y)$ is a diffeomorphism from $\mathcal{V}$ onto itself and for which $g$ satisfies for any $(x, y) \in \mathcal{B} \times \mathcal{V}$:

$$g(x, y) = g(x_0, \tau(x, y)) + C(x).$$

For simplicity, we write $\tau_x : y \mapsto \tau(x, y)$ by an abuse of notations. We know, by Corollary 1, that $x \mapsto \tau_x^{-1}(y_0)$ is continuous and converges to $\tau_{x_0}^{-1}(y_0) = y_0$. Hence, by restricting $x$ to a smaller neighborhood $\mathcal{B}' \subset \mathcal{B}$, we can ensure that $\tau_x^{-1}(y_0)$ belongs to $B(y_0, R')$ with $R'$ small enough so that $B(y_0, 2R') \subset \mathcal{V}$. Consider now the family of diffeomorphisms $\mathcal{P}$

$$\mathcal{P} = \{\mathcal{V} \ni y \mapsto \tau(x, y) \in \mathcal{V} | x \in \mathcal{B}'\}.$$

We have constructed $\mathcal{P}$ satisfying the conditions of Proposition 11. Moreover, by Proposition 1, the function $h(y) := g(x_0, y)$ satisfies a Łojasiewicz inequality in an open neighborhood $\mathcal{V}'$ of $y_0$:

$$\mu|h(y) - G(x_0)| \le \|\partial_y h(y)\|^2, \forall y \in \mathcal{V}'.$$

We can always choose the neighborhood $\mathcal{V}'$ to be an open ball $B(y_0, R)$ of radius $R > 0$ centered in $y_0$. Therefore, we have shown so far that $h$ and $\mathcal{P}$ satisfy the conditions of Proposition 11.

For any $\tau \in \mathcal{P}$, consider the ODE:

$$z_t = -\partial_y g(x, \tau(z_t)), \qquad z_0 \in B(y_0, R').$$

Following the notation in Proposition 11, we define $y_\tau := \tau^{-1}(y_0)$ for any $\tau \in \mathcal{P}$. We apply Proposition 11 which ensures stability of $z_t$. More precisely, there exists a positive constant $C$ smaller than $R'$ so that for any $0 < \epsilon < C$, the solution $z_t$ is well-defined at all times and satisfies $\|z_t - y_\tau\| \le \epsilon$ for any $t \ge 0$, provided that the initial condition $z_0$ satisfies $\|z_0 - y_\tau\| \le \eta$ for some positive $\eta < \frac{\epsilon}{2}$ that is independent of the choice of $\tau$ is $\mathcal{P}$:

$$\forall \tau \in \mathcal{P} : \|z_0 - y_\tau\| \le \eta \implies \|z_t - y_\tau\| \le \epsilon. \tag{22}$$

We will apply this result to a particular choice for $z_0$. From now on, we fix $0 < \epsilon < C$ and let $0 < \eta \le \frac{\epsilon}{2}$ be as in Proposition 11. Using Proposition 10, we know that $\phi_t(x_0, y)$ converges to $y_0 = \phi(x_0, y)$, hence there exits $T > 0$ s.t. $\|\phi_T(x_0, y) - y_0\| \le \frac{\eta}{3}$. Moreover, since the maps $x \mapsto \phi_T(x, y)$ and $x \mapsto y_{\tau_x}$ are continuous at $x_0$ with $y_{\tau_{x_0}} = y_0$, there exits $\eta'$ satisfying $0 < \eta'$ such that $B(x_0, \eta') \subset \mathcal{B}'$ and $\|\phi_T(x, y) - \phi_T(x_0, y)\| \le \frac{\eta}{3}$ and $\|y_{\tau_x} - y_0\| \le \frac{\eta}{3}$ for any $x \in B(x_0, \eta')$. Therefore:

$$\|\phi_T(x, y) - y_{\tau_x}\| \le \|\phi_T(x, y) - \phi_T(x_0, y)\| + \|\phi_T(x_0, y) - y_0\| + \|y_0 - y_{\tau_x}\| \le \eta.$$

For any $x \in B(x_0, \eta')$, by choosing $z_0 = \phi_T(x, y)$, we have that $\|z_0 - y_{\tau_x}\| \le \eta$. Therefore, we deduce by (22) that $\|z_t - y_{\tau_x}\| \le \epsilon$ and subsequently that

$$\|z_t - y_0\| \le \|z_t - y_{\tau_x}\| + \|y_{\tau_x} - y_0\| \le \epsilon + \frac{\eta}{3} \le \frac{7}{6}\epsilon,$$

since we imposed that $\eta < \frac{\epsilon}{2}$. Recall now that $z_t$ satisfies the ODE:

$$\dot{z}_t = -\partial_y g(x_0, \tau_x(z_t))\partial_z \tau_x(z_t).$$

By definition of $\tau_x$, we have $g(x, y) = g(x_0, \tau_x(y))$ for any $y \in B(y_0, 2R') \subset \mathcal{V}$. In particular, as we have shown that $\|z_t - y_0\| \leq \frac{7}{6}\epsilon < 2R'$, it follows that $z_t$ satisfies the ODE:

$$\dot{z}_t = -\partial_y g(x, z_t) = -\partial_y g(x_0, \tau_x(z_t))\partial_z \tau_x(z_t).$$

By Cauchy-Lipschtz theorem, the solution of the above ODE is unique. Moreover, since we know that $\phi_{T+t}(x, y)$ is a solution to the above ODE, then we deduce that $z_t = \phi_{T+t}(x, y)$. We have shown that for any $\epsilon < C$, there exists $T > 0$ and $\eta'$ such that:

$$\|\phi_t(x, y) - y_0\| \leq \frac{7}{6}\epsilon, \qquad \forall t \geq T, \quad \forall x \in B(x_0, \eta'). \tag{23}$$

Since $\phi_t(x, y)$ converges towards $\phi(x, y)$ by Proposition 10, taking the limit $t \to \infty$ in (23), we obtain:

$$\|\phi(x, y) - y_0\| \leq \frac{7}{6}\epsilon, \forall x \in B(x_0, \eta').$$

The above inequality imply in particular that $x \mapsto \phi(x, y)$ is continuous at $x_0$. $\qquad\square$

**Proposition 13** (**Stability of local minimizers**). *Let $x_0 \in \mathcal{X}$ and $y \in \mathcal{Y}$ and $g$ be such that Assumptions 1 to 3 hold. Assume that $y_0 = \phi(x_0, y)$ is a local minimizer of $y \mapsto g(x_0, y)$. Then, for any $x_1$ in a neighborhood of $x_0$, $\phi(x_1, y)$ is a local minimizer of $g(x_1, .)$.*

*Proof.* By assumption, $y_0 := \phi(x_0, y)$ is a local minimizer of $g(x_0, .)$ ensuring that $\partial_{yy}^2 g(x_0, y_0)$ is positive semi-definite. Moreover, by Corollary 1, there exists a neighborhood $B(x_0, \eta) \times B(y_0, 2R')$ of $(x_0, y_0)$ such that for any augmented critical point $(x_1, y_1) \in \mathcal{M} \cap B(x_0, \eta) \times B(y_0, 2R')$, the Hessian $\partial_{yy}^2 g(x_1, y_1)$ is similar to $\partial_{yy}^2 g(x_0, y_0)$. Hence, for any $(x_1, y_1) \in \mathcal{M} \cap B(x_0, \eta) \times B(y_0, 2R')$, $\partial_{yy}^2 g(x_1, y_1)$ must be positive semi-definite so that $y_1$ is a local minimizer of $g(x_1, .)$.

We can then apply Proposition 12 which ensures that $x \mapsto \phi(x, y)$ is continuous at $x_0$. Therefore, there exists $\eta' < \eta$ so that, for any $x_1 \in B(x_0, \eta')$, $y_1 := \phi(x_1, y)$ belongs to $B(y_0, 2R')$. As a result, $y_1$ must be a local minimizer of $g(x_1, .)$ since the augmented critical point $(x_1, y_1)$ belongs to $\mathcal{M} \cap B(x_0, \eta) \times B(y_0, 2R')$. $\qquad\square$

### B.4 Uniform Convergence of the Gradient Flow

The result bellow shows that the gradient flow $\phi_t(x, y)$ converges locally uniformly in $x$ near $x_0$ at an exponential rate, whenever $\phi(x_0, y)$ is a local minimum. It relies on the locally uniform convergence result in Proposition 12 and the locally uniform Łojasiewicz inequality in Proposition 1.

**Proposition 14.** *Let $x_0 \in \mathcal{X}$ and $y \in \mathcal{Y}$ and $g$ be such that Assumptions 1 to 3 hold. Assume that $y_0 := \phi(x_0, y)$ is a local minimum. Then there exists positive constants $\eta$, $T$, $\mu$ and $C$ such that:*

$$\|\phi_t(x, y) - \phi(x, y)\| \leq Ce^{-t\mu}, \qquad \forall t \geq T, x \in B(x_0, \eta).$$

*A fortiori, $\phi(x, y)$ is continuous on $B(x_0, \eta)$.*

*Proof.* Proposition 1 ensures the existence of $\epsilon > 0$ and $\eta > 0$ be such that the following inequality holds:

$$\mu|g(x, y') - G(x)| \leq \frac{1}{2}\|\partial_y g(x, y')\|^2, \qquad \forall x, y' \in B(x_0, \eta') \times B(y_0, \epsilon). \tag{24}$$

By Proposition 12 and for $\epsilon > 0$ small enough, there exists $T > 0$ and $\eta' > \eta > 0$ for which:

$$\|\phi_t(x, y) - y_0\| \leq \epsilon, \qquad \forall t \geq T, \quad \forall x \in B(x_0, \eta').$$

Therefore, choosing $y' = \phi_t(x, y)$ in (24) implies:

$$\mu|g(x, \phi_t(x, y)) - G(x)| \leq \frac{1}{2}\|\partial_y g(x, \phi_t(x, y))\|^2, \qquad \forall x \in B(x_0, \eta). \tag{25}$$

Note that $G(x)$ is the common value of $g(x, y)$ when $y$ is a critical point of $g(x, .)$ in $B(y_0, \epsilon)$. In particular, since $\phi(x, y) \in B(y_0, \epsilon)$, it holds that $G(x) = g(x, \phi(x, y))$. Moreover, by Proposition 13, $\phi(x, y)$ is a local minimum for $y \mapsto g(x, y)$. Hence, we must have $g(x, \phi_t(x, y)) - G(x) \geq 0$. We

may assume that the inequality is strict otherwise the $\phi_t(x, y)$ would be a fixed point and we would have $\phi_t(x, y) = \phi(x, y)$. The following inequality holds for any $t \geq T$:

$$\|\phi_t(x, y) - \phi(x, y)\| \leq \int_t^{+\infty} \|\partial_y g(x, \phi_s(x, y))\| \, \mathrm{d}s \leq -\frac{2}{\mu} \int_t^{+\infty} \dot{H}(s) \, \mathrm{d}s = \frac{2}{\mu} H(t).$$

where we introduced $H(t) = (g(x, \phi_t(x, y)) - G(x))^{\frac{1}{2}}$. Thus, we only need to study the evolution of $H(t)$ in time. Computing the derivatives of $H(t)$ and using the inequality in (25) yields

$$\dot{H}(t) = -\frac{1}{2} H(t)^{-1} \|\partial_y g(x, \phi_t(x, y))\|^2 \leq -\mu H(t).$$

By integrating the above inequality, it follows that $H(t) \leq H(T)e^{-\mu(t-T)}$. Moreover, using the smoothness of $y \mapsto g(x, y)$, we know that

$$H(T) \leq \sqrt{\frac{L}{2}} \|\phi_T(x, y) - \phi(x, y)\| \leq \sqrt{\frac{L}{2}} \epsilon, \forall x \in B(x_0, \eta).$$

Finally, we have shown that $\|\phi_t(x, y) - \phi(x, y)\| \leq \sqrt{\frac{L}{\mu}} \epsilon e^{-(t-T)\mu}$ for any $x \in B(x_0, \eta)$ and $t \geq T$. Since $\phi_t$ are continuous in $x$ and converge uniformly in $x$ on $B(x_0, \eta)$, then their limit must be continuous on $B(x_0, \eta)$. $\square$

## C   Differentiability of the Flow Selection

In this section, we study the differentiability of $x \mapsto \phi(x, y)$ through the evolution of $\partial_x \phi_t(x, y)$. The following result establishes that $\partial_x \phi_t(x, y)$ is well-defined and satisfies a linear differential equation.

**Proposition 15.** *Assume $g$ is twice continuously differentiable and satisfies Assumption 2. Then, $(x, t) \mapsto \phi_t(x, y)$ is continuously differentiable with $\partial_x \phi_t(x, y) := U_t(x, y)$ satisfying the differential equation:*

$$\dot{U}_t(x, y) = -B_t(x, y) - A_t(x, y)U_t(x, y), \tag{26}$$

*where $B_t$ and $A_t$ are given by:*

$$B_t(x, y) = \partial_{xy}^2 g(x, \phi_t(x, y)), \qquad A_t(x, y) = \partial_{yy}^2 g(x, \phi_t(x, y)).$$

*Proof.* The differentiability of the flow $\phi_t(x, y)$ in $x$ follows by the application of Cauchy-Lipschitz theorem. It suffices to differentiate the equation defining the flow w.r.t. to obtain (26). $\square$

Note that, by Proposition 10 and continuity of $\partial_{xy}^2 g(x, y)$ and $\partial_{yy}^2 g(x, y)$, the matrices $A_t(x, y)$ and $B_t(x, y)$ must converge to the following matrices $A_\infty$ and $B_\infty$ for any $(x, y) \in \mathcal{X} \times \mathcal{Y}$:

$$A_\infty(x, y) := \partial_{yy}^2 g(x, \phi(x, y)), \qquad B_\infty(x, y) := \partial_{xy}^2 g(x, \phi(x, y))$$

The following proposition shows that the pseudo-inverse of $A_\infty(x, y)$ remain bounded near $x_0$ provided that $\phi(x_0, y)$ is a local minimum.

**Proposition 16.** *Let $(x_0, y)$ be in $\mathcal{X} \times \mathcal{Y}$ and set $y_0$ and $g$ be such that Assumptions 1 to 3 hold. Assume that $y_0 := \phi(x_0, y)$ is a local minimum of $g(x_0, .)$. Then there exists an open neighborhood $\mathcal{U}$ of $x_0$ and a positive constant $\lambda > 0$ such that:*

$$\lambda \|A_\infty(x, y)\|_{op} \leq 1, \qquad \forall x \in \mathcal{U},$$

*where $A_\infty(x, y) = \partial_{yy}^2 g(x, \phi(x, y))$.*

*Proof.* We apply Proposition 9 which ensures the existence of an open neighborhood $\mathcal{V}$ of $(x_0, y_0)$ for which:

$$\partial_{yy}^2 g(x, y) = P(x, y)^\top \partial_{yy}^2 g(x_0, y_0) P(x, y).$$

where $(x, y) \mapsto P(x, y)$ is continuous map with values in $\mathbb{R}^{d \times d}$, and $P(x, y)$ is invertible for any $(x, y) \in \mathcal{V}$ with singular values in $[\sigma_{\min}, \sigma_{\max}]$ for $\sigma_{\min} > 0$ and $\sigma_{\max} < +\infty$. Moreover, since

$y_0 := \phi(x_0, y)$ is a local minimum of $g(x_0, .)$, we know, by Proposition 12, that $x \mapsto \phi(x, y)$ is continuous at $x_0$. Hence, there exists a neighborhood $\mathcal{U}$ of $x_0$ for which $(x, \phi(x, y)) \in \mathcal{V}$ for any $x \in \mathcal{U}$. Therefore, it follows that:

$$A_\infty(x, y) = P(x, \phi(x, y))^\top \partial_{yy}^2 g(x_0, y_0) P(x, \phi(x, y)), \qquad \forall x \in \mathcal{U}.$$

In particular, it follows that:

$$A_\infty(x, y)^\dagger = P(x, \phi(x, y))^{-1} \partial_{yy}^2 g(x_0, y_0)^\dagger P(x, \phi(x, y))^{-\top}.$$

Hence, we easily deduce that the operator norm of $A_\infty(x, y)^\dagger$ satisfies:

$$\left\| A_\infty(x, y)^\dagger \right\|_{op} \leq \sigma_{\min}^{-2} \left\| \partial_{yy}^2 g(x_0, y_0)^\dagger \right\|_{op}.$$

The result follows by setting $\lambda = \sigma_{\min}^2 \left\| \partial_{yy}^2 g(x_0, y_0)^\dagger \right\|_{op}^{-1}$. $\qquad\qquad\square$

We will need to introduce the following matrix $U^\star(x, y)$ defined as:

$$U^\star(x, y) := -(A_\infty(x, y))^\dagger B_\infty(x, y).$$

The following proposition shows, under mild conditions, that $U_t(x, y)$ converges towards a limiting element $U_\infty(x, y)$ satisfying the equation: $A_\infty U_\infty = A_\infty U^\star$.

**Proposition 17.** *Let $(x_0, y)$ be in $\mathcal{X} \times \mathcal{Y}$ and set $y_0$ and $g$ be such that Assumptions 1 to 3 hold. Assume that $y_0 := \phi(x_0, y)$ is a local minimum of $g(x_0, .)$. Then there exists $\eta > 0$ such that, for any $x \in B(x_0, \eta)$, $U_t(x, y)$ converges towards an element $U_\infty(x, y)$ satisfying*

$$A_\infty(x, y) U_\infty(x, y) = A_\infty(x, y) U^\star(x, y).$$

*In paticular, if $y$ is a critical point of $y \mapsto g(x, .)$ then $U_\infty(x, y) := U^\star(x, y)$. Moreover, there exists a time $T > 0$ and constants $C > 0$, $\mu$ such that for any $x \in B(x_0, \eta)$ and $t \geq T$:*

$$\|U_t(x, y) - U_\infty(x, y)\| \leq Ce^{-\mu t},$$

*Proof.* For simplicity, we omit the dependence on $(x, y)$ as they remain fixed. Let $P$ be a projection matrix that commutes with $A_\infty$, i.e. : $PA_\infty = A_\infty P$. We will choose $P$ to be either $P_\infty = A_\infty A_\infty^\dagger$ or $P = I - P_\infty$. Define $V_t = P(U_t - U^\star)$. By differentiating in time, it is easy to see that $\dot{V}_t$ satisfies:

$$\dot{V}_t = \tilde{B}_t - PA_t V_t. \tag{27}$$

where $\tilde{B}_t := P(B_\infty - B_t + (A_\infty - A_t)U^\star)$. Denote by $(s, t) \mapsto R_s^t$ the resolvent of the linear system (27), i.e. the squared matrix satisfying $\frac{dR_s^t}{dt} = -PA_t R_s^t$ for $t \geq s$ and $R_s^s = I$. Standard results for linear differential equations [45, Chapter 2] ensure that $R_s^t$ is always invertible at any time and that $V_t$ can be expressed in terms of $R_s^t$ as follows:

$$V_t = -R_0^t PU^\star + \int_0^t R_s^t \tilde{B}_s \, ds.$$

**Controlling $\|R_s^t\|_{op}$:**

We will show the following inequality:

$$\log\left(\left\|R_s^t\right\|_{op}\right) \leq \int_s^t \left(-\lambda_P + \|A_\infty - A_u\|\right) du, \tag{28}$$

where $\|.\|_{op}$ refers to the operator norm and $\lambda_P$ is the smallest eigenvalue of $PA_\infty P$. To achieve this, we define $\mathcal{L}_t = \frac{1}{2}\|R_s^t u\|^2$ for $t \geq s$ and $u$ a vector in $\mathcal{Y}$. We then differentiate $\mathcal{L}_t$ in time to get:

$$\begin{aligned}
\dot{\mathcal{L}}_t &= -\langle R_s^t u, PA_t R_s^t u\rangle = -\langle R_s^t u, PA_\infty R_s^t u\rangle + \langle R_s^t u, P(A_\infty - A_t)R_s^t u\rangle, \\
&= -\langle R_s^t u, PA_\infty P R_s^t u\rangle + \langle R_s^t u, P(A_\infty - A_t)R_s^t u\rangle, \\
&\leq 2\left(-\lambda_P + \|A_\infty - A_t\|_{op}\right)\mathcal{L}_t,
\end{aligned}$$

where we used that $PA_\infty P = P^2 A_\infty = PA_\infty$ since $P$ and $A_\infty$ commute. We also used elementary properties of the trace of product of matrices to get the last inequality. By integrating the above inequality, we obtain:

$$\frac{1}{2}\big\|R_s^t u\big\|^2 \mathcal{L}_t \le \frac{1}{2}\big\|R_s^s u\big\|^2 e^{2\int_s^t \left(-\lambda_P + \|A_\infty - A_u\|_{op}\right)\mathrm{d}u},$$

$$\le \frac{1}{2}\|u\|^2 e^{2\int_s^t \left(-\lambda_P + \|A_\infty - A_u\|_{op}\right)\mathrm{d}u},$$

where we used that $R_s^s = I$. The desired bound on $\|R_s^t\|_{op}$ follows by taking the supremum over $u$ in the unit ball.

**Controlling $\|B_\infty - B_t\|_{op}$ and $\|A_\infty - A_t\|_{op}$:**

By Proposition 14, there exists $\eta > 0$ and $T > 0$ such that:

$$\|\phi_t(x, y) - \phi(x, y)\| \le C e^{-t\mu}, \qquad \forall t \ge T, x \in B(x_0, \eta).$$

Moreover, since $\phi(x, y)$ is continuous at $x_0$ by Proposition 12, we can always choose $\eta$ small enough so that $\phi(x, y)$ remains bounded. Hence, there exists a compact set $K$ containing $\phi_t(x, y)$ for any $t \ge T$ and $x \in B(x_0, \eta)$. Denote by $|K|$ its diameter. By continuity of $\phi_t(x, y)$, we can also take $K$ large enough so that $\phi_t(x, y) \in K$ for any $0 \le t \le T$ and $x \in B(x_0, \eta)$. Since $g$ is three-times continuity differentiable by Assumption 1, there exists a positive constant $L$ s.t. for all $x \in B(x_0, \eta)$ and $y, y' \in K$:

$$\big\|\partial_{xy}^2 g(x, y) - \partial_{xy}^2 g(x, y')\big\|_{op}, \le L\|y - y'\|,$$

$$\big\|\partial_{yy}^2 g(x, y) - \partial_{yy}^2 g(x, y')\big\|_{op} \le L\|y - y'\|.$$

As a result, we can write

$$\max\left(\|B_\infty - B_t\|_{op}, \|A_\infty - A_t\|_{op}\right) \le L\|\phi(x, y) - \phi_t(x, y)\| \le c_t. \qquad (29)$$

where, we defined $c_t$ to be:

$$c_t = \begin{cases} LCe^{-t\mu}, & t \ge T, \\ 2L|K|, & t < T. \end{cases}$$

**Controlling $V_t$:** For simplicity define $C_t = \int_0^t c_u \, \mathrm{d}u \le C_\infty := 2L|K|T + LCe^{-T\mu}/\mu$. We will first control the error term $\int_0^t R_s^t \tilde{B}_s \, \mathrm{d}s$. For $t > T$, the following holds:

$$\int_0^t \big\|R_s^t \tilde{B}_s\big\|_{op} \mathrm{d}s \le \int_0^t \big\|R_s^t\big\|_{op}\left(\|B_\infty - B_t\| + \|A_\infty - A_t\|\|U^\star\|_\infty\right)\mathrm{d}s, \qquad (30)$$

$$\le \int_0^t e^{\int_s^t -\lambda_P + c_u \, \mathrm{d}u}\left(1 + \|U\|_{op}^\star\right)c_s \, \mathrm{d}s,$$

$$\le e^{C_\infty}\left(1 + \|U\|_{op}^\star\right)\int_0^t c_s e^{-(t-s)\lambda_P} \, \mathrm{d}s,$$

where we used elementary linear algebra inequalities for the first line and (28) and (29) for the second line. We need to control $\|U^\star(x, y)\|_{op} = \big\|A_\infty(x, y)^\dagger B_\infty(x, y)\big\|_{op}$. To achieve this, we use Proposition 16 which ensures that $\big\|A_\infty(x, y)^\dagger\big\|_{op} \le \lambda^{-1}$ for some positive $\lambda$ provided $x$ is close enough to $x_0$. Thus, we can choose $\eta$ small enough so that $\big\|A_\infty(x, y)^\dagger\big\|_{op} \le \lambda^{-1}$ for any $x \in B(x_0, \eta)$. Moreover, by Proposition 14, we know that $x \mapsto \phi(x, y)$ is continuous on $B(x_0, \eta)$ provided $\eta$ is small enough. Therefore, we can ensure that $B_\infty(x, y)$ is bounded by some value $B_{max}$ on $B(x_0, \eta)$. Hence, we deduce that $\|U^\star(x, y)\|_{op} \le M = \lambda^{-1}B_{\max}$ for any $x \in B(x_0, \eta)$. We can finally write the upper-bound bellow:

$$\int_0^t \big\|R_s^t \tilde{B}_s\big\|_{op} \mathrm{d}s \le e^{C_\infty}(1 + M)\underbrace{\int_0^t c_s e^{-(t-s)\lambda_P} \, \mathrm{d}s}_{E_t}. \qquad (31)$$

We distinguish two cases depending on the choice of $P$:

- **Case** $P = A_\infty A_\infty^\dagger$.

In the case where $A_\infty(x_0, y)=0$, then by Proposition 9 and for $\eta > 0$ small enough, it holds that $A_\infty(x, y) = 0$ for any $x \in B(x_0, \eta)$. In this case, the dynamics is trivial. Instead, if $A_\infty(x_0, y)\neq 0$, then by Proposition 9 and for $\eta > 0$ small enough, $A_\infty(x, y) \neq 0$ for any $x \in B(x_0, \eta)$. In this case, we know that $\|A_\infty(x, y)\|_{op}$ is the inverse of the smallest positive eigenvalue of $A_\infty(x, y)$ which is also equal to $\lambda_P$ by definition. Moreover, by Proposition 16, there exists $\eta > 0$ small enough and $\lambda > 0$ such that $\lambda\|A_\infty(x, y)\|_{op} \leq 1$ for any $x \in B(x_0, \eta)$. We then deduce that $\lambda < \lambda_P$. Hence, for $t \geq T$, we have:

$$E_t = c_T \int_0^T e^{-\lambda(t-s)} + LC \int_T^t e^{-\lambda(t-s)-(s-T)\mu},$$

$$= \frac{c_T}{\lambda} e^{-\lambda(t-T)} + \frac{LC}{\lambda - \mu} \left( e^{-\mu(t-T)} - e^{-\lambda(t-T)} \right).$$

By abuse of notation, we still write $\frac{1}{\lambda-\mu}\left(e^{-\mu(t-T)} - e^{-\lambda(t-T)}\right)$ even when when $\lambda = \mu$, to refer to the limit $(t - T)e^{-\lambda(t-T)}$ when $\mu$ approaches $\lambda$. By introducing $\tilde{\mu} = \frac{1}{2}\min(\lambda, \mu)$, we get the simpler bound:

$$E_t \leq \frac{(c_T + LC)}{\tilde{\mu}} e^{-\tilde{\mu}(t-T)}.$$

On the other hand, recalling the upper-bound on $\|R_s^t\|_{op}$ we deduce that $\|R_0^t\|_{op} \leq e^{C_\infty - \lambda_P t}$. Hence, we can write for any $t \geq T$:

$$\|V_t\| \leq e^{C_\infty}(1 + M)\left( e^{-\lambda_P t} + \frac{c_T + LC}{\tilde{\mu}} e^{-\tilde{\mu}(t-T)} \right),$$

$$\leq e^{C_\infty}(1 + M)\left( 1 + \frac{c_T + LC}{\tilde{\mu}} e^{\tilde{\mu}T} \right) e^{-\tilde{\mu}t}.$$

Hence, $V_t$ converges towards 0 at an exponential rate.

- **Case** $P = I - A_\infty A_\infty^\dagger$.

In this case, $\lambda_P = 0$ and $PU^\star = -PA_\infty^\dagger B_\infty = 0$. Therefore, $V_t$ simplifies to $V_t = \int_0^t R_s^t \tilde{B}_s \, \mathrm{d}s$. We will simply show that such integral is absolutely convergent. To achieve this, we consider $t \geq T$ and compute $E_t$:

$$E_t = \int_0^t c_s \, \mathrm{d}s = \int_0^T c_s \, \mathrm{d}s + \int_T^t c_s \, \mathrm{d}s,$$

$$= Tc_T + LC \int_T^t e^{-\mu(s-T)} \, \mathrm{d}s,$$

$$= Tc_T + \frac{LC}{\mu}\left( 1 - e^{-\mu(t-T)} \right) \leq Tc_T + \frac{LC}{\mu} := E_\infty.$$

Hence, $E_t$ converges to a finite quantity $E_\infty$. Using (31), we deduce that $\int_0^t R_s^t \tilde{B}_s \, \mathrm{d}s$ is absolutely convergent so that $V_t$ converges to an element $V_\infty$. Moreover, we have:

$$\|V_t - V_\infty\| \leq \int_t^\infty \left\| R_s^t \tilde{B}_s \right\| \mathrm{d}s \leq e^{C_\infty}(1 + M)(E_\infty - E_t),$$

$$\leq Ce^{C_\infty}\frac{L}{\mu}(1 + M)e^{-\mu(t-T)}.$$

Hence, we have shown that there exists $\eta > 0$ small enough such that for any $x \in B(x_0, \eta)$, $U_t(x, y)$ converges to an element $U_\infty(x, y)$ satisfying $A_\infty(x, y)U_\infty(x, y) = A_\infty(x, y)U^\star(x, y)$. Moreover, the there exists a time $T$ and positive constants $C'$ and $\mu'$ such that:

$$\|U_t(x, y) - U_\infty(x, y)\| \leq C'e^{-\mu't}, \forall t \geq T, \forall x \in B(x_0, \eta).$$

$\square$

*Proof of Theorem 2.* By Proposition 10, we have that $\phi_t(x,y)$ converges to $\phi(x,y)$. Moreover, since $\phi(x_0,y)$ is a local minimizer of $g(x_0,.)$, Proposition 12 ensures that $\phi(x,y)$ is continuous at $x_0$. Finally, we know by Proposition 15 that $\phi_t(x,y)$ is differentiable in $x$ and by Proposition 17 that $\partial_x\phi_t(x,y) := U_t(x,y)$ converges uniformly towards $U_\infty(x,y)$. Therefore, by [46, Theorem 7.17], we conclude that $\phi(x,y)$ is differentiable in a neighborhood of $x_0$ with differential given by $\partial_x\phi(x,y) = U_\infty(x,y)$. If in addition, $y$ is a local minimizer, then, by Proposition 17, $\partial_x\phi(x,y) = -\partial_{xy}g(x,y)(\partial_{yy}g(x,y))^\dagger$. $\qquad\square$

# D  Limits Points of Bilevel Optimization Algorithms

**Proposition 18.** *Let $g$ be a real-valued function on $\mathcal{X}\times\mathcal{Y}$ such that Assumption 2 holds. Consider the maps $\varphi_T$ and $\mathcal{I}_M$ defined in (33) and let $T$ and $M$ be non-negative integers, such that $T+M > 0$. Let $(x,y)\in\mathcal{X}\times\mathcal{Y}$ such that $\varphi_T(x,\mathcal{I}_M(x,y)) = y$. Then, $\partial_y g(x,y) = 0$.*

*Proof.* Let us fix $(x,y)\in\mathcal{X}\times\mathcal{Y}$ and consider the iterates $y^T = \varphi_T(x,y)$. We will show that $y^T$ satisfy a sufficient decrease condition for some positive constant $a$:

$$g(x,y^{T+1}) + \frac{L}{2}\|y^T - y^{T+1}\|^2 \le g(x,y^T). \tag{32}$$

To see this, we can use the smoothness of $g$ to write:

$$g(x,y^{T+1}) - g(x,y^T) \le -d^\top H_T d + \frac{L}{2}\|H_T d\|^2,$$

where $d = \partial_y g(x,y^T)$ and we write $H_T = H_T(x,y)$ by abuse of notation. Hence, it follows that:

$$g(x,y^{T+1}) - g(x,y^T) + \frac{L}{2}\|y^{T+1} - y^T\|^2 \le -d^\top\big(H_T - LH_T^2\big)d \le 0,$$

where we used that $H_T \le \frac{1}{L}I$. Similarly, we obtain a sufficient decrease condition for the iterates defined by $\mathcal{I}_M$. Consider now $T$ and $M$, such that $T+M > 0$, and let $(x,y)$ be such that $\varphi_T(x,\mathcal{I}_M(x,y)) = y$. Consider the iterates $y^k = \mathcal{I}_k(x,y)$ for $m \le M$, and $y^k = \varphi_t(x,y^M)$ for $t \le T$. Then the iterates $y^k$ define a non-increasing sequence $g(x,y^k)$. Moreover, since $y^{T+M} = y^0 = y$, it must be that $g(x,y^k) = g(x,y)$. The sufficient decrease condition in (32) implies that the iterates are all constant $y^k = y^0$. In particular, if $M > 0$, this implies that $H_M(x,y)\partial_y g(x,y) = 0$ so that $\partial_y g(x,y) = 0$ since $H_M(x,y)$ is invertible. On the other hand, if $M = 0$, then the condition $T+M > 0$ implies that $T > 0$, so that $y = y^1 = y - H_T(x,y)\partial_y g(x,y)$. Similarly, since $H_T(x,y)$, we deduce that $\partial_y g(x,y) = 0$. $\qquad\square$

**Proposition 19 (Properties of the maps $\varphi_T$ and $\mathcal{I}_M$).** *Let $g$ be a function satisfying Assumption 2 with a smoothness constant $L$. Consider $\varphi_T(x,y)$ and $\mathcal{I}_M(x,y)$ defined by the following recursion which holds for any $x,y\in\mathcal{X}\times\mathcal{Y}$:*

$$\varphi_{T+1}(x,y) = \varphi_T(x,y) - H_T(x,y)(\partial_y g(x,\varphi_T(x,y))), \qquad \varphi_0(x,y) = y \tag{33}$$
$$\mathcal{I}_{M+1}(x,y) = \mathcal{I}_M(x,y) - H'_M(x,y)(\partial_y g(x,\mathcal{I}_M(x,y))), \qquad \mathcal{I}_0(x,y) = y,$$

*where $H_T(x,y)$ and $H'_M(x,y)$ are positive symmetric matrices satisfying $H'_M(x,y) \le \frac{1}{L}I$ and $H_T(x,y) \le \frac{1}{L}I$ for any $(x,y)\in\mathcal{X}\times\mathcal{Y}$ and non-negative integers $T,M$. Moreover, assume that $H_T(x,y)$ is continuously differentiable. Then $\varphi_T$ and $\mathcal{I}_M$ satisfy Assumption 4.*

*Proof.* It is clear that for any $(x,y)\in\mathcal{X}\times\mathcal{Y}$, s.t. $y$ is a critical point of $g(x,.)$, we have that $\mathcal{I}_M(x,y) = \varphi_T(x,y) = y$. Moreover, if $T,M$ are such that $T+M > 0$ and $(x,y)\in\mathcal{X}\times\mathcal{Y}$ satisfy $\varphi_T(x,\mathcal{I}_M(x,y)) = y$, then Proposition 18 ensures that $\partial_y g(x,y) = 0$. It remains to obtain an expression for $\partial_x\varphi_T$ and $\partial_y\varphi_T$ in terms of second-order derivatives of $g$. We proceed by recursion. For $T = 0$, by setting $D = 0$, we have that:

$$\partial_x\varphi_0(x,y) = 0 = \partial^2_{xy}g(x,y)D, \qquad \partial_y\varphi_0(x,y) = I = I + \partial^2_{xy}g(x,y)D.$$

Let $(x,y)$ be an augmented critical point of $g$. Assume now that for some $T \ge 0$, there exists a matrix $D_T$, such that:

$$\partial_x\varphi_T(x,y) = \partial^2_{xy}g(x,y)D_T, \qquad \partial_y\varphi_T(x,y) = I + \partial^2_{yy}g(x,y)D_T. \tag{34}$$

Differentiating the expression of $\varphi_{T+1}(x,y)$ w.r.t. $x$ and $y$ yields:

$$\partial_x \varphi_{T+1}(x,y) = \partial_x \varphi_T(x,y) - \big(\partial_{xy}g(x,\varphi_T(x,y)) + \partial_x \varphi_T(x,y)\partial_{yy}^2 g(x,\varphi_T(x,y))\big)H_T(x,y)$$
$$- \partial_x H_T(x,y)\partial_y g(x,\varphi_T(x,y)).$$
$$= \partial_{xy}g(x,y)\big(D_T - \big(I + D_T\partial_{yy}^2 g(x,y)\big)H_T(x,y)\big) = \partial_{xy}g(x,y)D_{T+1},$$

Where we defined $D_{T+1}(x,y) = D_T - \big(I + D_T\partial_{yy}^2 g(x,y)\big)H_T(x,y)$. In the above expression, the last line follows by recalling that $\varphi_T(x,y){=}0$ and $\partial_x H_T(x,y)\partial_y g(x,\varphi_T(x,y)) = 0$ since $(x,y)$ is an augmented critical point of $g$ and by using the recursion assumption on $\partial_x \varphi_T(x,y)$.

Similarly, for $\partial_y \varphi_{T+1}(x,y)$, the following holds:

$$\partial_y \varphi_{T+1}(x,y) = \partial_y \varphi_T(x,y) - \partial_y \varphi_T(x,y)\partial_{yy}^2 g(x,\varphi_T(x,y))H_T(x,y)$$
$$- \partial_y H_T(x,y)\partial_y g(x,\varphi_T(x,y)),$$
$$= I + \partial_{yy}^2 g(x,y)\big(D_T - \big(I + D_T\partial_{yy}^2 g(x,y)\big)H_T(x,y)\big) = I + \partial_{yy}^2 g(x,y)D_{T+1}.$$

Hence, by recursion, $\varphi_T(x,y)$ satisfies the equation (34) for any $T \geq 0$. We have shown that $\varphi_T$ and $\mathcal{I}_M$ satisfy Assumption 4. $\qquad\square$

*Proof of Proposition 3.* Fix $T \geq$ and consider the iterates $(x_k, y_k)$ of Algorithm 1 using $\varphi_T$. By assumption $(x_k, y_k)_{k\geq 0}$ converges to an element $(x_T^\star, y_T^\star)$ in $\mathcal{X} \times \mathcal{Y}$. By continuity of the maps $\varphi_T$, $\mathcal{I}_M$ and $\partial_x \mathcal{L}_T$, we have that:

$$y_T^\star = \lim_k y_k = \lim_k \varphi_T(x_{k-1}, \mathcal{I}_M(x_{k-1}, y_{k-1})) = \varphi_T(x_T^\star, \mathcal{I}_M(x_T^\star, y_T^\star)),$$
$$\lim_k \partial_x \mathcal{L}_T(x_{k-1}, \mathcal{I}_M(x_{k-1}, y_{k-1})) = \partial_x \mathcal{L}_T(x_T^\star, \mathcal{I}_M(x_T^\star, y_T^\star)) := d^\star.$$

By Assumption 4, the first equation implies that $y_T^\star$ is a critical point of $g(x_T^\star, .)$ (i.e. $\partial_y g(x_T^\star, y_T^\star) = 0$). Moreover, taking the limit in the update equation $x_k = x_{k-1} - \gamma d_k$ yields $d^\star = 0$. Hence, we also have that $\partial_x \mathcal{L}_T(x_T^\star, \mathcal{I}_M(x_T^\star, y_T^\star)) = 0$. Finally, recall that $\mathcal{I}_M(x_T^\star, y_T^\star){=}y_T^\star$ by Assumption 4 since $(x_T^\star, y_T^\star)$ is an augmented critical point of $g$. Thus we have shown that:

$$\partial_x \mathcal{L}_T(x_T^\star, y_T^\star) = 0, \qquad \partial_y g(x_T^\star, y_T^\star).$$

Assume now that $y_T^\star$ is a local minimum of $g(x_T^\star, .)$ and that $(x_T^\star, y_T^\star)_{T\geq 0}$ is bounded. Hence, there exists a subsequence of $(x_T^\star, y_T^\star)_{T\geq 0}$ converging towards an accumulation point $(x^\star, y^\star)$. By abuse of notation, we denote $(x_T^\star, y_T^\star)_{T\geq 0}$ such subsequence. By continuity of the Hessian of $g$, it follows that $y^\star$ must also be a local minimum of $g(x^\star, .)$. We can now use Assumption 5 which ensures that $\varphi_T$ converges to a selection $\phi$. Moreover, since $\partial_x \varphi_T$ converges uniformly near local minima, it follows by [46, Theorem 7.17] that $\phi(x,y)$ is differentiable w.r.t. $x$ near $(x^\star, y^\star)$ and that $\partial_x \varphi_T(x,y)$ converges uniformly near $(x^\star, y^\star)$ towards $\partial_x \phi(x,y)$. Hence, we can write for $T$ large enough:

$$\partial_x \mathcal{L}_\phi(x_T^\star, y_T^\star) = \partial_x \mathcal{L}_T(x_T^\star, y_T^\star) + (\partial_x \phi(x_T^\star, y_T^\star) - \partial_x \varphi_T(x_T^\star, y_T^\star))\partial_y f(x_T^\star, y_T^\star),$$
$$= (\partial_x \phi(x_T^\star, y_T^\star) - \partial_x \varphi_T(x_T^\star, y_T^\star))\partial_y f(x_T^\star, y_T^\star).$$

By uniform convergence of $\partial_x \varphi_T(x,y)$ to $\partial_x \phi(x,y)$ and recalling that $(x_T^\star, y_T^\star)$ is bounded, we deduce that $\|\partial_x \mathcal{L}_\phi(x_T^\star, y_T^\star)\|$ converges to 0. In particular, this holds true for a subsequence satisfying $\limsup_T \|\partial_x \mathcal{L}_\phi(x_T^\star, y_T^\star)\| = \lim_T \|\partial_x \mathcal{L}_\phi(x_T^\star, y_T^\star)\|$, which proves the desired result. $\qquad\square$

*Proof of Proposition 4.* Let $(x,y) \in \mathcal{X} \times \mathcal{Y}$ be such that $y$ is a local minimum of $g(x,.)$. Define $d$ to be:

$$d = \partial_x \mathcal{L}_T(x,y) - \partial_{xy}^2 g(x,y)\big(\partial_{yy}^2 g(x,y)\big)^\dagger \partial_y \mathcal{L}_T(x,y).$$

By Theorem 2, $x \mapsto \phi(x,y)$ is differentiable at $x$ and since $y$ is a critical point of $g(x,.)$, the differential of $\phi(x,y)$ is given by $\partial_x \phi(x,y) = -\partial_{xy}^2 g(x,y)\big(\partial_{yy}^2 g(x,y)\big)^\dagger$. Hence, $d$ is equal to:

$$d = \partial_x \mathcal{L}_T(x,y) + \partial_x \phi(x,y)\partial_y \mathcal{L}_T(x,y).$$

Using the definition of $\mathcal{L}_T$ and recalling that $\varphi_T$ satisfies Assumption 4, the following holds:

$$
\begin{aligned}
d &= \partial_x f(x, \varphi_T(x, y)) + \partial_x \varphi_T(x, y) \partial_y f(x, \varphi_T(x, y)) + \partial_x \phi(x, y) \partial_y \varphi_T(x, y) \partial_y f(x, \varphi_T(x, y)), \\
&= \partial_x f(x, y) + (\partial_x \varphi_T(x, y) + \partial_x \phi(x, y) \partial_y \varphi_T(x, y)) \partial_y f(x, y), \\
&= \partial_x f(x, y) + (\partial^2_{xy} g(x, y) D + \partial_x \phi(x, y)(I + \partial^2_{yy} g(x, y) D)) \partial_y f(x, y), \\
&= \partial_x f(x, y) + \partial_x \phi(x, y) \partial_y f(x, y) + (\partial^2_{xy} g(x, y) + \partial_x \phi(x, y) \partial^2_{yy} g(x, y)) D \partial_y f(x, y), \\
&= \partial_x \mathcal{L}_\phi(x, y) + (\partial^2_{xy} g(x, y) + \partial_x \phi(x, y) \partial^2_{yy} g(x, y)) D \partial_y f(x, y).
\end{aligned}
$$

The last term of the above equation vanishes, since by definition of $\partial_x \phi(x, y)$, it holds that $\partial^2_{xy} g(x, y) + \partial_x \phi(x, y) \partial^2_{yy} g(x, y) = 0$. Therefore, we have shown that $d = \partial_x \mathcal{L}_\phi(x, y)$, which concludes the proof. $\qquad\square$

*Proof of Proposition 5.* By continuity of the maps $\varphi_T$ and $\mathcal{I}_M$ and since $(x_k, y_k, z_k) \to (x^\star, y^\star, z^\star)$, it holds that $y^\star = \varphi_T(x^\star, \mathcal{I}_M(x^\star, y^\star))$. Hence, by Assumption 4, it follows that $y^\star$ must be a critical point of $y \mapsto g(x^\star, y^\star)$, i.e. $\partial_y g(x^\star, y^\star) = 0$. Moreover, we have that $\tilde{y}_k = \mathcal{I}_M(x_k, y_k) \underset{k}{\to} \mathcal{I}_M(x^\star, y^\star) = y^\star$ by continuity of $\mathcal{I}_M$ and the condition in Assumption 4. Since $f$ and $\varphi_T$ are continuously differentiable we get that $u_k, v_k \underset{k}{\to} \partial_x \mathcal{L}_T(x^\star, y^\star), \partial_y \mathcal{L}_T(x^\star, y^\star)$. Moreover, recalling that $\mathcal{L}_T(x, y) = f(x, \varphi_T(x, y))$, by application of the chain rule and using that $\varphi_T(x^\star, y^\star) = y^\star$ it follows that:

$$
u_k \underset{k}{\to} u^\star := \partial_x f(x^\star, y^\star) + \partial_x \varphi_T(x^\star, y^\star) \partial_y f(x^\star, y^\star),
$$

$$
v_k \underset{k}{\to} v^\star := \partial_y \varphi_T(x^\star, y^\star) \partial_y f(x^\star, y^\star).
$$

By continuity of the higher-order derivatives of $g$, it holds that:

$$
\partial^2_{yy} g(x_k, y_{k+1}) \underset{k}{\to} A^* := \partial^2_{yy} g(x^\star, y^\star),
$$

$$
\partial^2_{xy} g(x_k, y_{k+1}) \underset{k}{\to} B^* := \partial^2_{xy} g(x^\star, y^\star).
$$

Recall that $z_k$ is given by the update equation $z_k = \mathcal{P}(A_k, v_k, z_{k-1})$, where $\mathcal{P}$ is a continuous map for which $z = \mathcal{P}(A, v, z)$ if and only if $z \in \arg\min_z \left\| A^2 z + v \right\|^2$. By continuity of $\mathcal{P}$, it follows that $z^\star$ satisfies:

$$
z^\star = \mathcal{P}(A^\star, v^\star, z^\star).
$$

Therefore, $z^\star$ minimizes $z \mapsto \left\| (A^\star)^2 z + v^\star \right\|^2$ and satisfies the fixed point equation $(A^\star)^3 z^\star + A^\star v^\star = 0$ so that $A^\star z^\star = -(A^\star)^\dagger v^\star$. Moreover, recall that $\xi_k = A_k z_k$, hence $\xi_k$ converges towards $\xi^\star := A^\star z^\star$. Therefore, $\xi^\star = -(A^\star)^\dagger v^\star$. Taking the limit as $k$ goes to $+\infty$, we get that $d_k$ defined in Algorithm 1 converges towards $d^\star$ defined by:

$$
\begin{aligned}
d^\star &:= u^\star + B^\star \xi^\star, \\
&= u^\star - B^\star (A^\star)^\dagger v^\star.
\end{aligned}
$$

By Proposition 4, it is easy to see that $d^\star = \partial_x \mathcal{L}_\phi(x^\star, y^\star)$. Finally, recalling the update equation $x_{k+1} = x_k - \gamma d_k$ and that $x_k \underset{k}{\to} x^\star$, we directly deduce that $d_k \underset{k}{\to} 0$, so that $d^\star = 0$. This shows that $(x^\star, y^\star)$ is an equilibrium point of (BGS) and satisfies (SC). $\qquad\square$

### D.1 Warm-start Strategy

In this section, we provide simple examples for the map $\mathcal{P}(A, v, z)$ to find approximate solutions minimizing $Q(z) := \frac{1}{2} \left\| A^2 z + v \right\|^2$, where $A$ is a symmetric matrix in $\mathbb{R}^{d \times d}$ satisfying $A \leq LI$, with $L$ being the smoothness constant of $g$ in Assumption 2. The algorithm $\mathcal{P}$ can be as simple as $N$-step of conjugate gradient descent on $Q$ with a step-size $\alpha \leq \frac{1}{L^4}$ where $L$ is the smoothness constant of $g$ in Assumption 2. More formally, $\mathcal{P}(A, v, z) = z^N$ where $z^N$ is the $N$ iterate of the following recursion:

$$
z^{n+1} = z^n - \alpha \partial_z Q(z^n), \qquad z^0 = z. \tag{35}
$$

| Algorithm | T | M | Correction |
|---|---|---|---|
| ITD [5] | $T > 0$ | $M = 0$ | False |
| Corrected ITD | $T > 0$ | $M = 0$ | True |
| Truncated ITD [47] | $T > 0$ | $M > 0$ | False |
| Corrected Truncated ITD | $T > 0$ | $M > 0$ | True |
| AID [42] | $T = 0$ | $T > 0$ | True |

Table 1: Recovering bilevel optimization algorithms from Algorithm 1.

It is clear that $\mathcal{P}(A, v, z)$ is continuous in its arguments. Moreover, using a similar argument as in Proposition 18, one can prove that whenever $z$ is a fixed point of $\mathcal{P}(A, v, z)$, then $z$ must be a critical point of $Q$ and therefore satisfies the equation $A^3 z + Av = 0$. The update equation in (35) depends however on the step-size $\alpha$ which needs to be smaller than $\frac{1}{L^4}$. to avoid the dependence on such step-size, A more efficient choice for the map $\mathcal{P}$ which does not require using a step-size, is to perform $N$ conjugate gradient iterations on $Q$ starting from an initial condition $z$.

### D.2 Recovering Existing Algorithms

Table 1 below summarizes how to recover well-known gradient-based algorithms for bilevel optimization from Algorithm 1. Hence, Algorithm 1 recovers the most popular bilevel optimization algorithms but also introduces a corrected version to them to ensure that they recover the equilibria of (BGS).

## E   Experiments

To illustrate the effect of the corrective term introduced in Section 5 , we consider two sets of experiments: a synthetic problem for which the optimal solutions can be computed in closed form and a dataset distillation task on Cifar10 [27] using a ResNet18 architecture [22].

### E.1 Synthetic Problem

Motivated by the instrumental variable regression problem [48] which solves a bilevel problem with quadratic objectives for both levels, we consider lower and upper-level objectives of the form:

$$f(x, y) := \frac{1}{2} x^\top A_f x + C_f^\top y$$
$$g(x, y) := \frac{1}{2} y^\top A_g y + y^\top B_g x$$

where $A_f$ and $A_g$ are symmetric positive matrices of size $d_x \times d_x$ and $d_y \times d_y$, $B_g$ is a $d_y \times d_x$ matrix and $C_f$ is a $d_y$ vector with $d_x$=2000 and $d_y$=1000. To allow for multiple solutions to the LL objective, we choose $A_g$ to be non-invertible with a null-space of dimension 100 while we choose $A_f$ to be invertible for simplicity. Furthermore, to ensure that $f$ admits a finite minimum value we choose $B_g$ to be of the form $A_g U$ for some randomly sampled matrix $U$. We construct the matrices $A_f$ and $A_g$ so that the highest eigenvalues of $A_f$ and $A_g$ are smaller than 1 and their conditioning is equal to 10. Here, we define the conditioning of a matrix to be the ratio between the highest and smallest non-zero eigenvalues. For a given $x$, the minimizers of $g$ are of the form:

$$y = -A_g^\dagger B_g x + (I - A_g A_g^\dagger) y_0,$$

where $y_0$ is any vector in $\mathbb{R}^{d_y}$. Replacing the optimal $y$ in the UL objective results in the expression which holds for any $y_0 \in \mathbb{R}^{d_y}$.

$$\frac{1}{2} x^\top A_f x - C_f^\top A_g^\dagger B_g x + C_f^\top (I - A_g A_g^\dagger) y_0.$$

At this point, it is easy to check that either maximizing or minimizing the above objective over $y_0$ results in an infinite value of the objective whenever $C_f^\top (I - A_g A_g^\dagger)$ is non-zero. This implies that the optimistic and pessimistic formulations of the bilevel problem result in an infinite optimal loss.

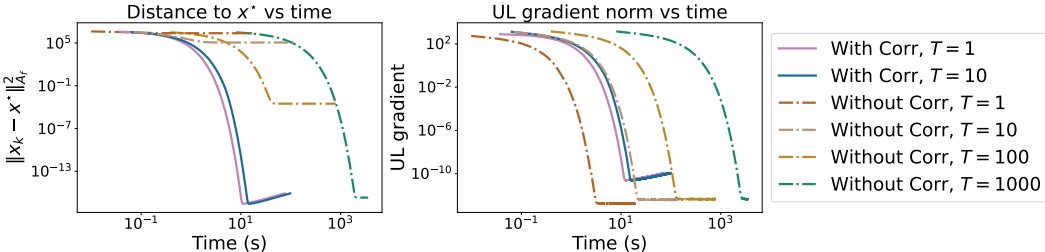

Figure 3: (left) Evolution of the distance of the UL iterate $x_k$ to the equilibrium $x^\star$ vs time (in seconds). (right) evolution of the norm of approximate gradient $d_k$ vs time in seconds. In all cases, algorithms are run until convergence, i.e. $\|d_k\|$ converges to 0.

However, the (BGS) has a well-defined solution. To see this, it is possible to define a selection of the form $\phi(x,y) = -A_g^\dagger B_g x + (I - A_g A_g^\dagger)y$ which corresponds to the limit of a gradient flow of $g$ initialized at $y$. The upper objective of (BGS) is therefore given by:

$$\mathcal{L}_\phi(x,y) = \frac{1}{2}x^\top A_f x - C_f^\top A_g^\dagger B_g x + C_f^\top (I - A_g A_g^\dagger)y.$$

Instead of optimizing $\mathcal{L}_\phi(x,y)$ over $x$ and $y$ which would result in an infinite loss, (BGS) optimizes $\mathcal{L}_\phi(x,y)$ over $x$ only, while $y$ is optimized for $f(x,y)$, thus seeking an equilibrium $(x^\star, y^\star)$ satisfying (SC) which can be expressed in closed form as

$$x^\star := A_f^{-1} B_g^\top A_g^\dagger C_f, \qquad y^\star := -A_g^\dagger B_g x + (I - A_g A_g^\dagger)y_0$$

where $y_0$ is any vector in $\mathbb{R}^{d_y}$. Hence, while there exist multiple equilibria, they all have the same value for $x^\star$ and yield a finite objective.

We solve the above problem using Algorithm 1 either using the correction or not. When using the correction, we compute the approximate solution $\xi_k$ to the linear system (5) using the following update rule:

$$\xi_k = \xi_{k-1} - \beta(\partial_{yy}g(x_{k-1}, y_k)\xi_{k-1} + v_k) \tag{36}$$

where $\beta = 0.9$ is a positive step-size. For the lower-level problem, we use $T$ steps of gradient descent with a step-size $\alpha = 0.9$ while we set the upper-level step-size to $\gamma = 1..$ We then set the warm-start parameter value $M$ to 0 and vary $T$.

**Results.** We consider the distance of the iterate $x_k$ to the optimal equilibrium $x^\star$ as measured by the metric induced by $A_f$:

$$\|x_k - x^\star\|_{A_f}^2 := \frac{1}{2}(x_k - x^\star)^\top A_f(x_k - x^\star)$$

Figure 3 (left) shows the evolution of $\|x_k - x^\star\|_{A_f}^2$ as a function of time (in seconds) for different algorithmic choices, while Figure 3(right) shows the evolution of the approximate upper-level gradient $d_k$ used in Algorithm 1. We first observe that, without correction, and when using a small number of unrolled iterations ($T \leq 10$), the algorithm does not converge towards $x^\star$, (the distance to the iterate is larger than $10^3$). Instead, the algorithm reaches a different equilibrium as suggested by the evolution of the gradient approximation $d_k$ towards 0 (Figure 3-(right)). As the number of unrolling steps $T$ increases, the algorithm takes more time to converge as suggested by Figure 3-(right) (green trace $T = 1000$). However, the limit gets closer to the equilibrium $x^\star$ (Figure 3-(right), green trace). This confirms our first convergence result in Proposition 3 stating that unrolled optimization finds an approximate solution to (BGS).

When using the correction, Algorithm 1 is able to recover the equilibrium $x^\star$ while still using a small number of unrolling steps $T \leq 10$ and requiring less time to converge. This observation supports the result in Proposition 5.

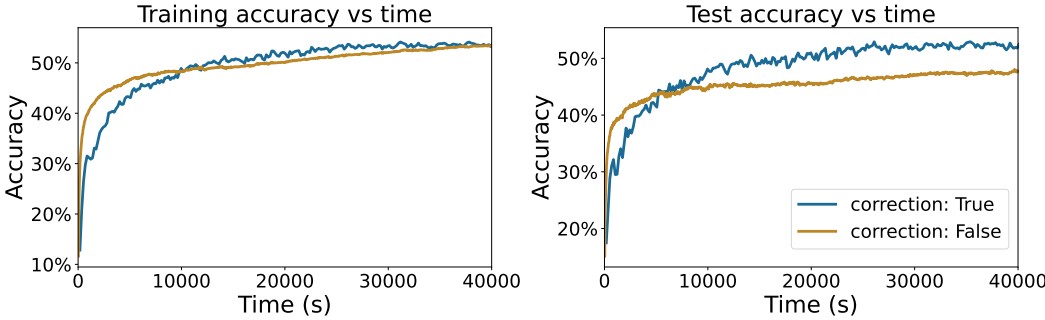

Figure 4: Evolution in time of the training and test accuracy of a ResNet18 model on Cifar10 dataset. Each iteration corresponds to the accuracy of the model with parameter $y_k$ trained on a synthetic dataset of 100 points $x_k$ to minimize the LL objective. The synthetic points $x_k$ are learned by minimizing the training error when using the running model $y_k$.

## E.2 Dataset Distillation on Cifar10

We consider the task of learning a small synthetic dataset so that a classifier trained on such a dataset achieves a small error on a training set. More formally, we consider a classification problem with $C$ classes using a model with parameters $y$ and a training dataset $\mathcal{D}_{tr} = \{(\xi_i, c_i)\}$ consisting of $N$ i.i.d. samples $\xi_i$ and corresponding labels $c_i$. The goal is to learn a synthetic dataset of $FC$ points, where $F$ is a positive integer, such that each class $c$ contains $F$ representative samples. We can collect the synthetic points into a vector $x$ to be learned and denote by $\mathcal{D}_x$ the synthetic dataset. For a given dataset $\mathcal{D}$, denote by $\mathcal{L}_\mathcal{D}(y)$ the cross-entropy loss of a model with parameters $y$ evaluated on $\mathcal{D}$. The bi-level formulation of the distillation task consists in optimizing a lower-level objective $g(x, y) = \mathcal{L}_{\mathcal{D}_x}(y)$ to learn the model parameters $y$ that best predicts the classes of the synthetic dataset. The upper-level objective $g(x, y) = \mathcal{L}_{\mathcal{D}_{tr}}(y)$ evaluates the optimal model on the training set and optimizes the synthetic samples.

**Setup**   . We consider a setup similar to [52] for distilling `Cifar10` [27] on 100 synthetic points. We set $F{=}10$, thus requiring 10 synthetic points for each of the $C{=}10$ classes of `Cifar10`. We then use ResNet18 [22] as a classifier and apply Algorithm 1 to learn the optimal synthetic points. For the lower level, we use gradient descent with 1 unrolled iteration (i.e. $T = 1$, $M = 0$) and a step-size of $\alpha{=}0.001$. For the upper level, we use Adam optimizer [26], with the default parameters, a step-size of $\gamma = 0.01$ and a batch-size of 1024. When using the corrective term, we use the update equation (36) with a step-size $\beta = 0.0001$.

**Results.**   Figure 4 shows the evolution of the training and test accuracy of the model as a function of time in two settings, either with or without correction. While the training accuracy for both versions of the algorithm is similar, the corrective term yields an improved final test accuracy ($54.19\%$ vs $48.6\%$). Note that these accuracies are of the same order as those obtained in [51] suggesting that distilling Cifar10 in only 100 samples is not sufficient to capture all variability in the dataset. While the additional correction increases the computational cost per iteration, it provides a better gradient estimate which results in a faster/better performance overall.