# OpenReview forum: "Non-Convex Bilevel Games with Critical Point Selection Maps"
_NeurIPS.cc/2022/Conference — NeurIPS 2022 Accept_

### Official Review · Reviewer_oofy · 2022-07-09

**Rating:** 2
**Confidence:** 4
**Soundness:** 2 fair
**Presentation:** 3 good
**Contribution:** 1 poor

**Summary:**

Bilevel optimization problems present an ambiguous model when the lower-level problem has multiple solutions. Typical ways to deal with this ambiguity are to consider either the so-called "optimistic" or "pessimistic" versions of the problem. In contrast, this paper proposes the use of a selection map to resolve the ambiguity. Under certain conditions, the authors show the selection map is differentiable which they claim makes the model amenable to numerical computation.

**Questions:**

What do the authors mean by their notation for the BGS problem? Should the $y$ variable in the selection map in the upper-level problem be interpreted as a kind of "initial condition" or "state variable"?

**Strengths And Weaknesses:**

The strengths of the paper are related to the technical rigor of the work. The results look reasonable and as would be expected under the specified conditions, though I did not check the proofs. The weaknesses of the paper are that the notation is unclear and, more significantly, that the relevance of the selection map model is questionable.

The authors present the pessimistic-BG problem using non-standard notation of separately presenting an upper-level and a lower-level problem. The typical notation is to write a single optimization problem where the lower-level problem is included as a constraint in the upper-level problem. The authors' notation is understandable to readers with familiarity with bilevel optimization, but their notation would be unclear to readers not familiar with bilevel optimization.

However, the notation that the authors use to present the selection map within the BGS problem is unclear. As currently written, the authors' notation is unclear, and the resulting model does not resolve the ambiguity that they seek to resolve. The authors' notation does resolve the ambiguity if an additional assumption, which is not explicitly stated in the paper, is made -- specifically that the $y$ variable within the selection map is a kind of "initial condition" or "state variable".

Regardless of the above issue about the meaning of how the selection map is used, there is questionable relevance of this model of using a selection map to resolve ambiguity in bilevel optimization problems with lower levels that have non-unique solutions. The reason this model is questionable is that depending on how this selection map is chosen the results of the model can be completely different, and so the selection map approach does not really resolve any ambiguity in which lower-level solution is used in the upper-level problem. The authors present no evidence that this would be a meaningful model. Even if the authors provide evidence that the model is meaningful, it is questionable that this represents an important or major contribution. Based on my understanding of the authors' unclear notation, their idea of using a selection map is equivalent to merely defining a different (parametric) objective function for the upper-level problem. As such, the authors' problem is equivalent to an optimistic bilevel optimization problem with a different (parametric) objective function. The technical results are also arguably: not surprising, not particularly difficult to derive or prove, and not interesting.

---

> ### Author Response · Authors · 2022-08-02
> **Response to Reviewer 4**
>
> Thank you for your comments. We hope the present discussion brings some clarifications.
>
> 1. **BGS is not an optimistic bilevel problem:** *“The authors' problem is equivalent to an optimistic bilevel optimization problem with a different (parametric) objective function.”*
>       - We believe we highlighted the differences between these two problems (see for instance intro, sec 3.1 and 3.2 and L151-156).
> Our formulation results in a game between the LL and UL agents and there is absolutely no incentive to select a solution $y$ that minimises the UL cost as done in optimistic BP.
>       - For a concrete example, please refer to the numerical study in Appendix E.1 where we discuss a problem  for which both optimistic and pessimistic formulations result in an infinite objective, (thus trying to solve it would yield a diverging solution) while the BGS formulation results in a finite value for which a solution can be computed.
>
>
> 2. **Quality of the technical results:**
> 	*“The technical results are also arguably: not surprising, not particularly difficult to derive or prove, and not interesting.”*
>
>     - We view the intuitive nature of the results as a strength since it establishes that ‘implicit differentiation’ still makes sense even when the solutions are degenerate and the implicit function theorem is not applicable (which is the case when using overparameterized models). This problem remained open until now,  as it was unclear how to proceed without the Implicit function theorem. Hence, regardless of the practical implications, these results are of independent theoretical interest. This result is also interesting for practitioners, (as pointed out by Reviewer 3 (tW4t)), as it shows that one can safely apply implicit differentiation to a large class of problems that arise in practice.
>
> 3. **The considered problem is relevant:**
>     - *“The authors present no evidence that this would be a meaningful model. Even if the authors provide evidence that the model is meaningful, it is questionable that this represents an important or major contribution.”*
>
>         - We believe our proposed framework to be **highly relevant** as it allows us to study the behaviour of many algorithms used in practice for solving bilevel problems (see Prop 3 and 5) in scenarios that often arise in practice including non-convex and  highly over-parameterized lower objective (see (Vicol et al. [47]))
>         - We also believe it to be a **major contribution** as it answers open questions such as: *“Does it make sense to use implicit differentiation even when the loss has multiple degenerate critical points and the implicit function theorem doesn’t hold?”*. Our answer is yes and one can even improve the quality of the result by doing so as shown by propositions 5 and illustrated in the numerical study in Appendix E.
>
>     - *“The reason this model is questionable is that depending on how this selection map is chosen the results of the model can be completely different”*
>    The proposed problem BGS formalises common practice for solving bilevel problems in ML where the lower loss is non-convex and possesses degenerate critical points (overparameterization). It shows that these algorithms are implicitly choosing a selection map and solving the corresponding BGS. While the choice of the selection results in a different BGS problem, we do not believe this fact to be a limitation but rather a strength as  it formalises the implicit bias of practical algorithms. Note also that we proposed a “canonical choice” for the selection in 4.2 as the limit of the continuous-time gradient flow and we showed that many practical algorithms solve a  BGS with such a selection.
>
> 4. **The selection resolves the ambiguity:**
> The proposed selection, as defined in Def 1, along with the definition of the BGS resolves the ambiguity in the bilevel problem without any additional condition. That is because, given a selection map, the BGS problem is a well-defined game. Any game is completely defined by two cost functions: here $g$ for the LL agent and $\mathcal{L}_{\phi}$ for the UL agent. The goal is to find an equilibrium point $(x^*, y^*)$ satisfying eq (SC). The interpretation of $y$ as an initial condition comes from the use of a selection arising from a gradient flow (in section 4), however, this interpretation is not necessary for defining BGS unambiguously.
>
>
> 5. **Clarity of the notation:**
>     - **Formulation of the pessimistic-BG:** We used a formulation that is consistent with the rest of the paper, but we are happy to amend it as suggested.
>     - **Notation for the selection:** we believe the notation to be clear as it is defined mathematically as a map $\phi(x,y)$ from $X\times Y$ to $Y$ that satisfies the conditions described in definition 1:  Criticality and self-consistency.  As we are not sure which part is unclear in this definition, and if you still have questions about this, please let us know, we’ll be happy to clarify it.

---

### Official Review · Reviewer_tW4t · 2022-07-11

**Rating:** 7
**Confidence:** 3
**Soundness:** 4 excellent
**Presentation:** 4 excellent
**Contribution:** 4 excellent

**Summary:**

The paper analyses bilevel optimization problems with a non-convex lower level problem through the concept of a selection map that chooses a particular solution to the lower problem. They use this notion to define a new selection map based on gradient flows and analyze the resulting games and the differentiability of the selection using a new set of tools in Morse theory. They finally use the analysis to propose an algorithm for differentiating through the inner problem even when the unrolled optimization of the inner problem is solved for a finite number of steps.

**Questions:**

I think my comments under the weaknesses section are addressable and would appreciate the authors making amends along those directions.

**Minor Comments**
Minor Typo : line 266 : knew -> known



**Limitations:**

Yes

**Strengths And Weaknesses:**

**Strengths**
1. The paper is well written and was comprehensible even for a practitioner like me without much prior knowledge about the specific theoretical tools used to analyze such problems. It also provided a nice overview of the existing approaches which I really appreciated.
2. As a practitioner, I think the primary contribution of the paper was showing that the rough equivalent of the implicit function theorem is actually the theoretically sound approach to take even in the case where the inner problem is non-convex (which is what we used to do even otherwise, but it’s very useful to actually have a theoretical justification for it ;) ) under reasonable assumption. To my knowledge this is the first paper showing this.
**Weaknesses**
1. Although, I realize that it’s probably unreasonable of me to expect experiments from a theory paper, I would have appreciated some experiments showing the practical impact of applying the gradient correction on some toy problems.

---

> ### Author Response · Authors · 2022-08-02
> **Response to Reviewer 3**
>
> Thank you very much for the encouraging comments. We are glad that you found the paper clear given the amount of theoretical and technical content. It is also heartwarming for us to read that you find the results useful as a theoretical justification for common practice. We fixed the typo in a revised version.
>
> **Experimental validation:** Thank you for helping us make a stronger point. As suggested, we performed numerical experiments, presented in Appendix E, to illustrate the impact of applying the gradient correction on two problems:
> - A synthetic problem for which the optimal solution is known. This problem allows us to clearly assess the performance of each algorithm and shows a clear benefit of using the gradient correction. It also illustrates the results of Prop 3 and Prop 5, that unrolling solves BGS as the number of unrolling steps increases and that the corrected algorithm does so without the need for increasing the unrolling steps.
> - A dataset distillation task on Cifar10 using a ResNet18 network. We observed a relative improvement of 11.5% in the test accuracy when using the correction.

---

### Official Review · Reviewer_YZdt · 2022-07-12

**Rating:** 6
**Confidence:** 4
**Soundness:** 3 good
**Presentation:** 2 fair
**Contribution:** 3 good

**Summary:**

This paper studies bilevel optimization. The authors show that existing algorithms is subject to approximation errors. The authors introduce a simple correction to these algorithms for removing the errors.

**Questions:**

1. It says that many existing algorithms for bilevel optimization suffer from approximation errors. But it only analyzes the "Unrolled Optimization Scheme"
2. Step 9 in Algorithm 1 is using approximation, which is not precise. How can one implement it?
3. Any concrete example to show the other algorithms issue would be much better.
4. It only solves the bilevel optimization to stationary point and is not able to solve it to global optimality.

**Ethics Review Area:**

["I don’t know"]

**Limitations:**

1. It says that many existing algorithms for bilevel optimization suffer from approximation errors. But it only analyzes the "Unrolled Optimization Scheme"
2. Step 9 in Algorithm 1 is using approximation, which is not precise. How can one implement it?
3. Any concrete example to show the other algorithms issue would be much better.
4. It only solves the bilevel optimization to stationary point and is not able to solve it to global optimality.

**Strengths And Weaknesses:**

Strength: A good background survey of bilevel optimization. The writing is reasonable.

Weakness: 1. It says that many existing algorithms for bilevel optimization suffer from approximation errors. But it only analyzes the "Unrolled Optimization Scheme"
2. Step 9 in Algorithm 1 is using approximation, which is not precise. How can one implement it?
3. Any concrete example to show the other algorithms issue would be much better.
4. It only solves the bilevel optimization to stationary point and is not able to solve it to global optimality.

---

> ### Author Response · Authors · 2022-08-02
> **Response to Reviewer 2**
>
> Thank you for your response and for the clarifying questions and for helping us make a stronger point.  We hope the following answers all your questions. Please also refer to Appendix E for numerical verification and to the general response for a discussion of the contributions. If you have any other questions or comments, we’ll be happy to discuss them further.
> 1. **Contributions of the paper:**
> As you pointed out,  one of the contributions of the paper is to show that some existing algorithms in bilevel optimization approximately solve the Bilevel Game that we introduced (BGS) and we show that a simple correction to these algorithms can remove a bias error. These contributions are consequences of two central contributions of this paper which we also summarized in the general response:
> 	- Introducing a new bilevel problem BGS and showing that it is the problem solved by commonly used bilevel algorithms when the LL has multiple solutions.
> 	- Introducing new theoretical tools for studying the differentiability of the problem, which is central for gradient-based algorithms. This is challenging since classical results (such as the implicit function theorem) do not hold and we had to develop new strategies for obtaining our results.
>
> 2. **What algorithms are covered?**
>       - In this work, we considered two classes of bilevel algorithms commonly used in ML:  ‘unrolled optimization’ and ‘implicit gradient'. Algorithm 1 covers both settings as we discuss in Table 1 depending on the choice of parameters $M$ and $T$:
>           - For $M{=}0$ and $T{>}0$ and no correction, we get ‘unrolled optimization’.
>           - For $M>0$, $T=0$ and with correction, we recover the ‘implicit gradient’.
>       - **“it only analyzes the "Unrolled Optimization Scheme" “:**  Thank you for pointing this out, we clarify in the revised version that the approximation errors occur only when using ‘unrolled optimization’ and not when using ‘implicit gradient methods’ by saying:
> “We leverage this characterization to show that popular algorithms based on Iterative Differentiation (ITD) admit fixed points that are approximately equal to the BGS's equilibrium points up to an error due to finite computational power.”
>
>       - The correction we propose is a way of incorporating the ‘implicit gradient’ to ‘unrolled optimization’ and has the effect of removing the approximation error as shown in Prop 5 and numerically in Appendix E.
>
> 3. **Practical implementation of the correction:**
> 	Thank you for pointing this out. Please refer to Appendix E for a practical implementation (eq 36) that we used for the numerical results. It consists of one or more steps of gradient descent to solve the linear system. Other possible choices are discussed in Appendix D.1. (eq 35). We will release an implementation of the algorithm upon acceptance of the paper.
>
> 4. **Concrete examples:**
> Please refer to Appendix E for numerical experiments and a to the second general response. The experiments show indeed a substantial improvement due to the corrective term in two settings: a synthetic one inspired by instrumental variable regression (Singh et al. 2019) with a ground truth solution to the problem and a dataset distillation task on Cifar10 using a ResNet18 network.
>
> 5. **Solving bilevel to stationarity instead of globally:**
> Indeed, we do not claim to find a global optimum in the non-convex case since the algorithm relies on gradient methods which are local algorithms and are thus only guaranteed to reach stationary points on non-convex problems.  Despite this fact, these algorithms seem to work well in practice even for bilevel optimization (Lorraine et. al.  2020). The situation is no different in our case. Thus, we do not believe this to be a limitation of the method, instead, this reflects the fact that solving a high dimensional non-convex problem is NP-hard in general. Note, however, that the algorithms would find a global solution whenever gradient descent is able to do so:  For instance, if the lower-level is strongly convex then the implicit differentiation ( in our case it corresponds to Algorithm 1 with correction,  $M{>}0$  and $T{=}0$)  can find a global inner-solution as shown in [2,19]. However, we consider here a much more challenging setting where the LL is non-convex and where gradient methods are only guaranteed to reach a stationary point which is not necessarily a global optimum.

---

### Official Review · Reviewer_gXUJ · 2022-07-19

**Rating:** 6
**Confidence:** 1
**Soundness:** 3 good
**Presentation:** 2 fair
**Contribution:** 3 good

**Summary:**

The paper studies the nonconvex inner problem bilevel optimization, proposed algorithm for such a problem, and theoretically show the asyptotically convergence.

**Questions:**

How to understand the UL in (BGS)? We also have the variable of $y$ and the minimization is only taken in $x$.

In Definition 1, 1.Criticality, is it $\partial_y g(x, \phi(x, y)) = 0$? Moreover, is self-consistency needed?


**Limitations:**

The paper lack numerical verifications, and I am doubt about the setting of selection map, my inituition is that it is a mapping from $\mathcal{X}$ to $\mathcal{Y}$.

**Strengths And Weaknesses:**

The paper provides the comprehensive study of the problem and the result is complete.

---

> ### Author Response · Authors · 2022-08-02
> **Response to Reviewer 1**
>
> Thank you for your response, for the clarifying questions, and for helping us make a stronger point. We hope the following answers all your questions. Please refer to Appendix E for numerical verification and to the general response for a discussion of the contributions. If you have any other questions or comments, we’ll be happy to discuss them further.
> 1. **Numerical verification:**
> Please refer to the second general response and to Appendix E for numerical experiments. The experiments show indeed a substantial improvement due to the corrective term in two settings: a synthetic one inspired by instrumental variable regression (Singh et al. 2019) with a ground truth solution to the problem (to allow precise assessment of the solution) and a dataset distillation task on Cifar10 using a ResNet18 network.
>
> 2.  **“Why define the selection map as a function from $X\times Y$ to $Y$ instead of simply a function from $X$ to $Y$?“**
>
>     - The dependence of $\phi(x,y)$ on a variable $y$ accounts for the warm-start of the inner variable which is common practice as it yields improved results for bilevel problems in an overparameterized regime (Vicol et al 2022, [47]). Hence, the problem we introduce, (BGS) formalizes these common practices and opens the way for a better theoretical understanding of those as well as practical improvements (such as the correction we propose, see also numerical results in appendix E).
>     - Intuitively, $y$ selects the basin of attraction of the function $g(x,y)$ and $\phi(x,y)$ is the critical point corresponding to such a basin. In the particular case when $g(x,y)$ admits a unique critical point (if, for instance, $g(x,y)$ is strongly convex in $y$)  the selection happens to be independent of $y$ and is equal to the unique minimizer $y^{\star}(x)$, i.e. $\phi(x,y)= y^{\star}(x)$.
>
> 3. **Why does the UL minimize over $x$ only and not $y$?**
>     - **Motivation:** The proposed formulation is motivated by the common practice when solving non-convex/overparameterized bilevel problems in ML which optimize $y$  only w.r.t. the LL objective and uses warm-start  (Vicol et al 2022, [47]). We show that BGS is the problem that these algorithms are trying to solve:  (Proposition 3 and 5).
>     - This work departs from existing standard problem formulations (such as pessimistic or optimistic problems) where the UL problem optimizes over both x and y. Instead, BGS proposes a game formulation, where UL optimizes only over x while the LL objective optimizes over y. The end goal becomes to obtain an $(x^{\star},y^{\star})$ satisfying stationarity equations for both UL and LL losses (eq  (SC) ). In short: the LL selects a point $y$ which uniquely determines a solution $\phi(x,y)$, then the UL optimizes $x$ using such solution  $\phi(x,y)$
>     - Qualitatively, this has major implications: For instance, in the synthetic experiment (Appendix E), the optimistic/pessimistic formulation (where UL optimizes over both x and y) results in an infinite objective. However, BGS results in finite objectives and solutions.
>     - Please note that the BGS problem recovers exactly the standard bilevel problem when the LL objective admits a unique solution as discussed in L148-150.
>
> 4. **Properties of the selection:**
>       - **Criticality:** it is indeed given by $\partial_y g(x,\phi(x,y))=0$, thank you for pointing this out, we fixed the typo in the revised version.
>       - **Self-consistency:**  It can be best understood when thinking of $\phi(x,y)$ as the limit of a gradient flow of $g(x,y)$ starting from $y$. In that case, we know that whenever y is a critical point, then the flow is constant which directly gives self-consistency. More intuitively, the selection provides a fixed point for any choice of $y$. In particular, if $y$ is already a fixed point, it is natural/economical to choose it, hence the condition $\phi(x,y)=y$.

---

> > ### Comment · Reviewer_gXUJ · 2022-08-07
> > **Thank you for the response**
> >
> > Thank you for the response. It solves my questions, however, I still adhere my original judgement.

---

### Author Response · Authors · 2022-08-02
**General Response**

Thank you all for your effort and for your time in reviewing our paper. We are glad that three reviewers lean towards acceptance and that Reviewer 3 highlights our theory results as providing a theoretically sound justification for using approximate implicit differentiation even when the implicit theorem does not hold, a situation that often occurs in practice.

Following the reviewers’ recommendations, we added **two numerical experiments in Appendix E** of the revised paper, to illustrate our results and demonstrate the positive effect of using the proposed correction. We discuss the results in a separate general response.  We do emphasize, however, that our contributions are theoretical and would like to provide some clarifications/context for the problem we consider:

**Context:**  Bi-level optimization when inner problems are non-convex are ill-defined. Yet, many strategies are used in practice such as "using the implicit function theorem as if the inner problem was strongly convex" or unrolled optimization with a fixed number or increasing number of steps.  Two natural questions arise: "how can we solve the ambiguity problem" and "are these practical heuristics justified? what do they actually optimize"?  These questions are thus highly relevant since the theory for these (successful) methods remains to be done.

**Solution to the ambiguity (section 3):**  To solve the ambiguity problem, we need a selection mechanism among critical points. The selection map does not have to be parametric or explicitly defined. It can be simply related to the implicit bias of the algorithm used in the inner loop. But, how do we formalize it?  What are the properties needed for these selection maps?  What is the corresponding optimization problem? This leads us to the definition of BGS games.

**Theoretical contributions (section 4):**  After proposing the BGS framework, comes the question of the existence of such selection maps and their differentiability, which will be needed to justify the use of existing algorithms.  We answer this point by considering selections that are limits of gradient flows and provide a positive answer from a continuous-time perspective. Differentiability of such limits is a difficult problem, and we found the class of parametric Morse-Bott functions to be the right analytical tools for this (those are slightly more restricted than functions satisfying the KL property, which have recently led to some breakthroughs in non-convex optimization).  Finding this framework is absolutely non-trivial, given the fact that we address problems that are largely open in the literature.

**Practical implications (section 5):**
1. A consequence of these results is that strategies using “the implicit function theorem” are still valid even though such theorem does not technically hold (because of possible degeneracy of the hessian).
2. These results allow us to prove that many algorithms used for bilevel optimization are actually solving a well-defined problem which is the BGS problem we introduced (Prop 3 and 5). In addition, we show that algorithms based on unrolled optimization can be ‘corrected’ using an AID like term so that they find solutions to BGS without the need for increasing the number of unrolling steps.

---

> ### Author Response · Authors · 2022-08-02
> **General response - Numerical results**
>
> Following the reviewers’ recommendations, we consider two sets of experiments presented in Appendix E of the revised paper:
> 1. A synthetic experiments inspired by the instrumental variable regression (Singh et al 2019) for which a closed form solution can be computed thus providing ground truth for comparison.
> 2. A dataset distillation task on Cifar10 using a ResNet18 architecture.
>
> All results show an improvement when using the proposed correction in Algorithm 1.
> 1. **Synthetic example:**
>       - Without correction, Unrolled optimization with a small number of steps ($T \leq 10$) does not converge towards the ground truth solution. Increasing the number of unrolling steps to $T=1000$ allows the algorithm to converge towards the ground truth as suggested by Prop 3. However, this comes at an increased computational cost as shown in Figure 1 Appendix E.
>       - With correction, the algorithm converges towards the ground truth even with a small number of unrolling ($T=1$ or $10$), thus confirming the result in Prop 5.  The algorithm is also faster overall despite the additional approximate linear solve (Figure 1 Appendix E).
>       - As discussed in appendix E, the optimistic/pessimistic formulation of this synthetic example results in an infinite loss, unlike the problem we consider (BGS) which results in finite objectives and solutions. This clarifies the statement of Reviewer 4 (oofy) about the connection between BGS and the optimistic bilevel problem.
> 2. **Distillation task:**
>       - The correction results in an improved test accuracy compared to no correction (Figure 2 of appendix E).
>
> These numerical results support our theoretical findings. We do emphasize, however, that our contributions are theoretical as discussed in the general response.

---

### Author Response · Authors · 2022-08-09
**Discussion with the reviewers**

Dear reviewers,

Thank you again for your time and effort in assessing our submission.

We also thank you reviewer (gXUJ) for your acknowledgment of the response and we are happy to see that we answered all your questions.

We understand that this is the summer break but we hope to hear from the rest of the reviewers so we get a chance of addressing any point that could need more clarification.

In particular, we hope to hear from reviewer (oofy)  whose low score, as we believe, is based on a misunderstanding that we hope we have clarified in the response.

Thank you again for your help

---

### Meta-Review · Area_Chair_v5Hg · 2022-08-26

**Recommendation:** Accept
**Confidence:** Less certain

**Metareview:**

This paper studies bilevel optimization problems and proposes techniques for disambiguating cases where the lower-level objective has multiple optimal solutions. The main points the reviewers raise in favor of acceptance are that:

1. The paper provides theoretical justification for techniques used to solve ambiguous bilevel optimization in practice, and this theory leads to algorithmic improvements.
1. The paper is well written.
1. The topic and results are of interest to the NeurIPS community.

Reviewer oofy argues strongly for reject with the following main concerns:

1. The notation of the paper is confusing (and in particular, there are concerns about the use of the variable y when the bilevel games with selection (BGS)). Reviewer gXUJ also had some confusion about the BGS setup, and reviewer tW4t agreed that the use of y is inconsistent in the BGS setup.
1. Introducing a selection map doesn’t really resolve ambiguity in the optimization problem because choosing the selection map is essentially resolving the ambiguity by hand.

In the rebuttal, the authors argue that the theoretical analysis is still interesting and useful because many algorithms used to solve bilevel optimization problems in practice are implicitly making a choice of selection map, and the theoretical analysis of the paper allows us to understand what those techniques are really optimizing for. Reviewer tW4t also found the example selection maps provided in the paper to be compelling. I am convinced by the author rebuttal and reviewer tW4t that the selection map is useful.

As for the notational concerns in the introduction of BGS, I think the paper would benefit from added discussion on the role of y in the upper level. In particular, from the author responses and later sections of the paper, it appears that y plays the role of a warm-start or initialization for the agent optimizing in the lower level. This discussion should be included near to the introduction of BGS, since otherwise it is confusing why the selection map is not a function from X -> Y.



**Award:**

No

---

### Decision · Program_Chairs · 2022-09-14

Accept